

# Revealing inflow and wake conditions of a 6MW floating turbine

Nikolas Angelou[1], Jakob Mann[1], and Camille Dubreuil-Boisclair[2]

[1]Technical University of Denmark, Department of Wind and Energy Systems, Frederiksborgvej 399, 4000, Roskilde, Denmark
[2]Equinor ASA, Sandslivegen 90, 5254, Sandsli, Norway

**Correspondence:** Nikolas Angelou (nang@dtu.dk)

**Abstract.** We investigate the characteristics of the inflow and the wake of a 6MW floating wind turbine from the *Hywind Scotland* offshore wind farm, the world's first floating wind farm. We use two commercial nacelle-mounted lidars to measure the up- and downwind conditions, with a fixed and a scanning measuring geometry, respectively. In the analysis, the effect of the surge and sway motion of the nacelle on the lidar measuring location is taken into account. The upwind conditions are

parameterized in terms of the mean horizontal wind vector at hub height, the shear and veer of the wind profile along the upper part of the rotor and the induction of the wind turbine rotor. The wake characteristics are studied in two narrow wind speed intervals $8.5 – 9.5 \ \mathrm{ms}^{-1}$ and $12.5 — 13.5 \ \mathrm{ms}^{-1}$, corresponding to below and above rotor rated speeds, respectively, and for turbulence intensity values between $3.3\% – 6.4\%$. The wake flow is measured by a wind lidar scanning in a horizontal plan position indicator mode, which reaches ten rotor diameters downwind. This study focuses on the downstream area between 3

and 8 rotor diameters. In this region, our observations show that the transverse profile of the wake can be adequately described by a self-similar wind speed deficit, that follows a Gaussian distribution. We find that even small variations ($\sim 1\% – 2\%$) of the ambient turbulence intensity can result in an up to 10% faster wake recovery. Furthermore, we do not observe any additional spread of the wake due to the motion of the floating wind turbine.

## 1 Introduction

The increase of the global renewable energy capacity is largely based on the continuation of the current expansion rate of the wind energy sector and the reduction of the associated production costs (Wiser et al., 2021). To accomplish this increase, wind turbines and wind farms will continue to increase in size, with a growing focus on offshore wind energy production. Despite the operational challenges that wind turbines encounter in offshore conditions, several factors favor offshore deployment: the size of the available areas, the generally good wind resource, and the spatial homogeneity of wind conditions. The latter and in

connection with the current foundation and the operation and maintenance costs further justify the increasing size of the wind turbines (Jørgensen et al., 2021), that form offshore wind farms.

    In general, to optimize the energy production of wind farms and to ensure that they are operational during their expected lifetime, it is necessary to be able to model realistically intra- and inter-farm flows, e.g., the individual wakes, the blockage in front and speed up around the wind farms, and the wind farm wake. In the case of intra-farm flows, the wind turbines

are typically located in the far wake of adjacent wind turbines, where the characteristics of the flow are mainly related to



the ambient wind conditions, which are usually described by the inflow speed, shear and veer profiles, turbulence intensity, atmospheric stability, and surface roughness. A thorough review of the current state-of-the-art physical description of those flows is presented in Porté-Agel et al. (2020), who highlights the importance of realistic physical parameterization of the flow in the far-wake to enable accurate prediction of the wind turbine production and the wind-induced loads. The impact

of atmospheric conditions on wake properties is investigated in numerical studies (e.g. Wu and Porté-Agel, 2012; Rodrigues et al., 2015) and wind tunnel experiments (e.g. Chamorro and Porté-Agel, 2009, 2010). The results of such studies have formed our current knowledge that, for example, an increase of turbulence intensity results to a faster wake recovery and that stable atmospheric stratification favours the downwind propagation of the wake trace.

Due to the varying sea depth of the exclusive economic zone of coastal countries and the wind farm construction restrictions

regarding the minimum distance from land, floating offshore wind farms will often be an attractive option. In the case of floating wind turbines, the wind turbine motion that results from the interaction of the floating structure with the wind and sea has to be considered, in addition to the atmospheric conditions. The wake characteristics of floating wind turbines have been studied using numerical (Wise and Bachynski, 2020; Kleine et al., 2021; Nanos et al., 2021; Chen et al., 2022; Li et al., 2022) and wind tunnel (Fu et al., 2019; Schliffke et al., 2020) experiments. Those studies focussed on the impact of the sway (Fu

et al., 2019; Nanos et al., 2021; Li et al., 2022), surge (Fu et al., 2019; Schliffke et al., 2020; Chen et al., 2022) and heave (Kleine et al., 2021) motions on the turbine operation. Yet, the absence of the study of the motion of floating turbine under realistic atmospheric turbulence conditions limits our understanding on how the impact of the motion of the wind turbine on the wake properties can be utilized in control strategies that optimize the operation of wind turbines and subsequently wind farms.

In this context the conduction of field campaigns is necessary for enhancing our knowledge about the physics that govern the wake physics. In this respect, the development of remote sensing sensors based on the light detection and ranging (lidar) technique has extended the study of wake flows generated by full-scale wind turbines in the atmospheric boundary layer. This was achieved since their design enables the acquisition of wind observations with high spatial and temporal resolution while operating either ground-based (e.g. Iungo et al., 2013; Iungo and Porté-Agel, 2014; Archer et al., 2019; Menke et al., 2020)

or nacelle-mounted (e.g. Bingöl et al., 2008; Aitken et al., 2014; Aitken and Lundquist, 2014; Carbajo Fuertes et al., 2018; Schneemann et al., 2021; Cañadillas et al., 2022; Brugger et al., 2022). Wind lidars are now used to detect the propagation of the wake trace (Archer et al., 2019) and, subsequently, measure both the mean (Iungo et al., 2013; Iungo and Porté-Agel, 2014; Menke et al., 2020) and the dynamic (Carbajo Fuertes et al., 2018) wake characteristics. Thus, they are used to evaluate the accuracy of different wake models (Trujillo et al., 2011; Trabucchi et al., 2017; Brugger et al., 2022). Wakes in the far region are

investigated using wind scanning wind lidars installed on both the ground and the nacelle (Archer et al., 2019; Carbajo Fuertes et al., 2018; Brugger et al., 2022).

In this study, we use two wind Doppler lidars installed on the nacelle of a floating wind turbine that is part of a small offshore wind farm with five wind turbines. Using two lidars enables the synchronous monitoring of the up- and down-wind conditions. Thus it was feasible first to acquire observations of the characteristics of the wake and subsequently relate those to

the inflow wind statistics. Similar studies using nacelle-mounted lidars have been performed on onshore wind turbines focusing





on wake model validation (Carbajo Fuertes et al., 2018) or investigating the relationship between the wake meandering and the fluctuations of the transverse wind component (Bingöl et al., 2008; Trujillo et al., 2011; Brugger et al., 2022).

Here we present, to our knowledge, the first combined inflow and wake measurements of a floating offshore wind turbine using two nacelle-mounted wind lidars. In Section 2, we describe the field campaign and the measuring configuration of the two wind lidars and the data post-processing steps. In Section 3, we present the two models used to describe the acquired wind lidar measurements in the up- and down-wind directions based on the characteristics of the free and wake flow. Finally, we present our results in Section 4 where the accuracy of both the up- and down-wind wind lidar is assessed, the properties of the mean free wind field are estimated and the wake flow characteristics are quantified for different inflow wind cases.

## 2 Field Study

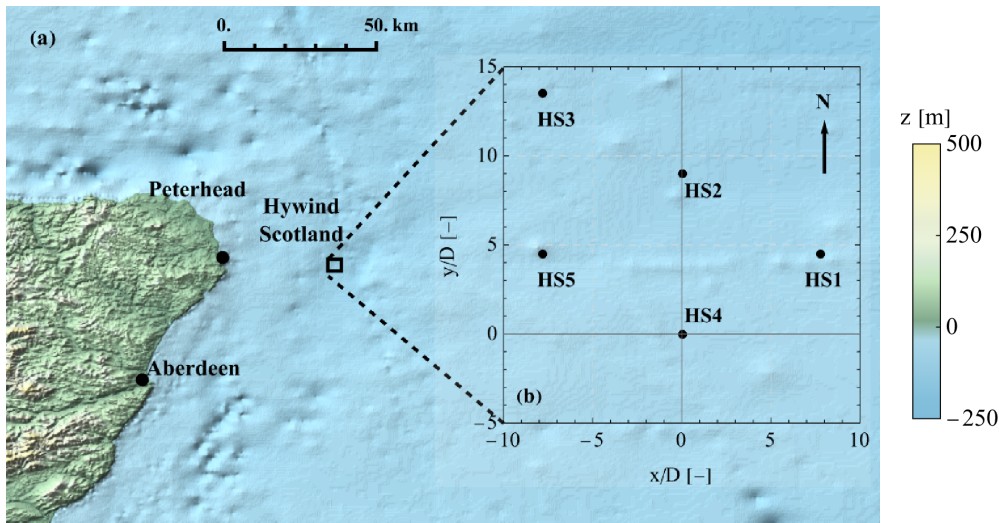

**Figure 1.** (a) Map of the eastern coast of Scotland where the offshore *Hywind Scotland* wind farm in located (black rectangle). (Inset, b) Drawing of the wind farm layout consisting of five wind turbines (denoted as HS1, HS2, HS3, HS4 and HS5) using a right-handed coordinate system whose $y$-axis is aligned to the North and the origin is placed at the location of the wind turbine (HS4).

*Hywind Scotland* is the world's first commercial floating offshore wind farm. It is located approximately 26 km from the east coast of Scotland (see Fig. 1(a)). In total, five Siemens Gamesa SWT-6.0-154 turbines with a hub height $H = 98.6$ m, a rotor diameter $D = 154$ m and a rotor rated speed of $10.1$ ms$^{-1}$ have been mounted on a ballast floating platform (Jacobsen and Godvik, 2021). The closest distance between neighbouring wind turbines is approximately equal to 1387 m ($9D$). The wind turbines' (denoted as HS1, HS2, HS3, HS4, and HS5) locations form a "W"-shape layout, with a clockwise rotation of $30°$ in relation to the North, as shown in Fig. 1(b). The layout is presented in a right-handed coordinate system whose $y$-axis is aligned to the North and the origin is the location of the HS4 wind turbine. The HS4 wind turbine was instrumented





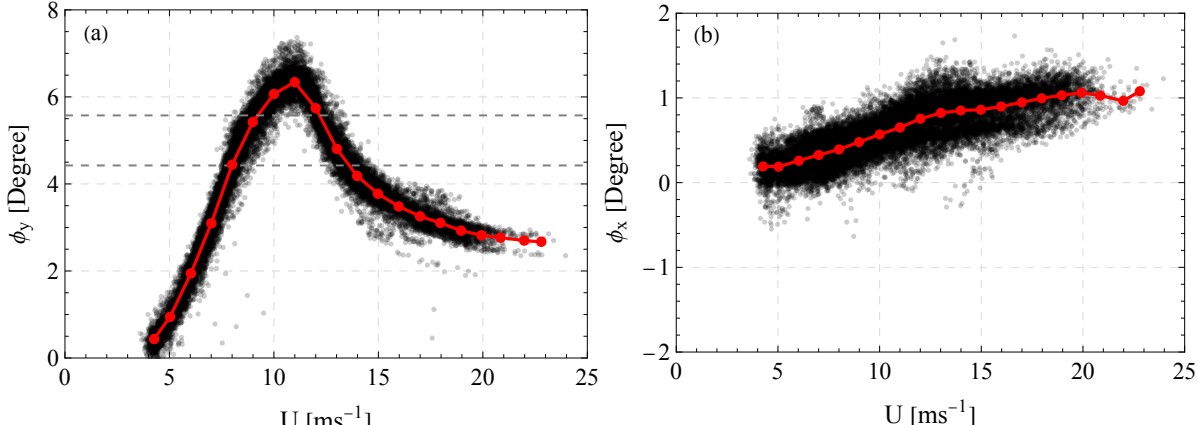

**Figure 2.** (a) Pitch $\phi_y$ and (b) roll $\phi_x$ values of the HS4 wind turbine for different wind inflow speeds $U$. The data presented in black correspond to the mean measurements of the MRU installed on the nacelle over a 2-minute period. The red line depicts the mean value of the pitch and roll angle per wind speed. The gray dashed lines in the pitch plot highlight the interval used in the analysis of this study (see more in Section 4.3).

with two nacelle-mounted wind lidars: a *Wind Iris* lidar by *Vaisala Oyj* (Finland) to measure the inflow and a *G4000 Galion* lidar by *SgurrEnergy* (Scotland) to measure the wake. The two wind lidars operated for a period of 3.5 months between June and September 2020. Furthermore, during this period, a *G4000 Galion* wind lidar was installed on the HS2 wind turbine to

measure downwind relative to that turbine. Both HS2 and HS4 wind turbines were additionally instrumented with a motion reference unit (MRU) to monitor the motion of the nacelle, by measuring the rotation about the longitudinal and transverse axes in relation to the yaw direction of the turbine. The data acquired from the MRU were logged by the supervisory control and data acquisition (SCADA) system at 1 Hz. Figure 2 presents the pitch $\phi_y$ and roll $\phi_x$ angles averaged over a period of approximately 2 minutes as a function of the mean horizontal wind speed $U$. The 2-minute period was selected to study the

response of the wind turbine over a period that was similar to the duration of the wake scan performed by the *G4000 Galion* wind lidar (see more in Section 2.1). The pitch corresponds to a rotation about the transverse axis relative to the yaw direction and the roll about the longitudinal. We observe that both the pitch and roll angles depend on the wind speed. The pitch angle has an increasing trend with wind speed, reaching approximately 7° at 11 ms$^{-1}$ (rotor rated speed), followed by a decrease down to 3°. Positive pitch angles indicate a counter-clockwise rotation of the rotor plane. The roll angle shows a constant and slightly

increasing trend with increasing wind speed, which on average varies between 0° and 1°, balancing the torque of the rotor. Jacobsen and Godvik (2021) studied the dynamic response of the wind turbines at *Hywind Scotland* and found an increase in the standard deviation of both the pitch and roll angles with an increasing wind speed. However, up to mean wind speeds of 20 ms$^{-1}$, those variations are normally less than 1°, indicating that the response of the floating platform is very stable.





## 2.1 Wind lidars

The two nacelle-mounted wind lidars used were the *Wind Iris* and the *G4000 Galion* monitoring the upwind and downwind conditions, respectively. The *Wind Iris* is a pulsed Doppler wind lidar. Doppler wind lidars measure the *radial speed* which corresponds to the projection of the wind vector on the line-of-sight of a laser beam. In the case of the *Wind Iris*, radial wind speed measurements are acquired over four fixed line-of-sights, which form the corners of a rectangle at any fixed distance. The radial wind speed sets were acquired at a sampling rate of 0.25 Hz in 10 separate range gates corresponding to the upwind

distances 50 m ($0.32D$), 80 m ($0.52D$), 120 m ($0.78D$), 160 m ($1.04D$), 200 m ($1.30D$), 240 m ($1.56D$), 280 m ($1.82D$), 320 m ($2.08D$), 360 m ($2.34D$), and 400 m ($2.60D$), when the *Wind Iris* was horizontally leveled. The first seven range gates were located within the area covered by the rotor, while the last three were located outside the rotor plane with respect to the transverse direction (see Fig. 3, red points). The four telescopes defining the geometry are enclosed in a single module installed 4.5 m above the rotor center, which is 103.4 m above sea level, and 4 m behind the rotor plane. Furthermore, the telescope of

the *Wind Iris* was tilted by 2.5° with respect to the horizontal plane to bring the center of symmetry of the measuring plane to the hub height at $2.5D$, when the wind turbine was not operating.

The *G4000 Galion* mounted on the HS4 wind turbine is also a pulsed Doppler lidar where a scanner controls the direction of the beam. This lidar was configured to do a Position Indicator (PPI) scan. The PPI scan consisted of 41 line-of-sight measurements spanned from $-20°$ to $20°$ in the azimuthal direction. Each line-of-sight measurement consisted of 50 range

gates that extended from 15 m to 1485 m ($9.6D$), sampled at 0.32 Hz. Furthermore, the PPI scan was performed with an elevation angle of 5°. This elevation angle was selected to compensate for the pitch angle of the wind turbine's nacelle so that the scan was horizontal when the mean wind speed was between either $8.0 – 9.5$ ms$^{-1}$ or $12.0 – 13.5$ ms$^{-1}$, see Fig. 2a. The pulsed scanning Doppler lidars have the advantage that they can acquire instantaneous measurements along a beam with a fixed spatial resolution. However, the required pulse accumulation time for producing reliable estimates of the radial speed limits the

scanning speed, especially when measurements at long ranges are required. For these reasons, the *G4000 Galion* wind lidar completed a PPI scan in approximately 2 minutes, following 26 seconds of resetting the scanner head to its initial position. Thus, approximately 12 scanning patterns were acquired per 30-minute period. The measurements were grouped in a Cartesian grid whose $x$-axis was aligned with the downwind direction parallel to the yaw direction of the wind turbine, with an origin at the center of the wind turbine's rotor. The grid consisted of three-dimensional cells with dimensions $D/5 \times D/5 \times D/5$. The

size of the cells was determined by the length of the range gates (30 m) and resulted in having at least one measurement per grid cell per scanning pattern (see Fig. 3).

The *Galion* on the HS2 wind turbine scanned in two alternating modes, i.e., a PPI and a Range Height Indicator (RHI) scan. The PPI scan had the same characteristics as the one performed by the *Galion* lidar on HS4. The RHI scan consisted of 36 line-of-sight measurements with elevation angles that extended from $-15°$ to 20°. In our study, the RHI measurements acquired

on the HS2 wind turbine were used only to investigate the average vertical motion of the wake (see Appendix A).

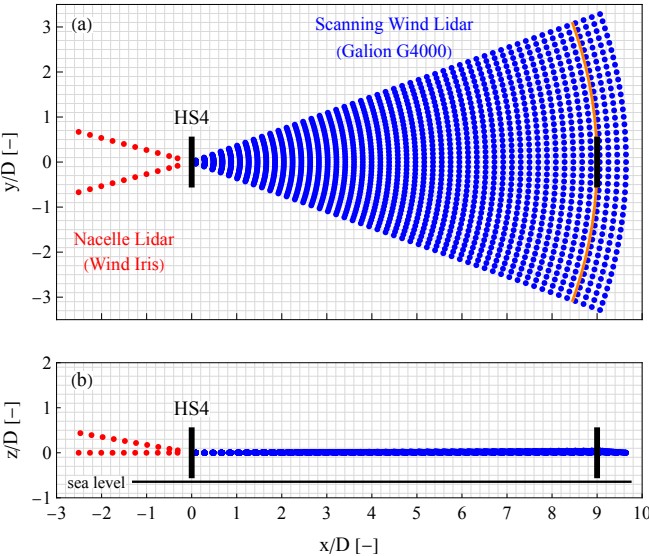

**Figure 3.** Top (a) and side (b) view of the measuring configuration used to monitor the up- and downwind flow relative to the HS4 wind turbine when the pitch of the rotor is equal to $5°$. The scanning pattern of the *Galion G4000* is presented with blue points and the measuring locations of the *Wind Iris* with red points. The coordinate system is defined such that the $x$-axis is aligned with the downwind direction relative to the wind turbine, and its origin is located at the wind turbine's rotor. The figure also depicts the expected position of the rotor of the HS1, HS2, and HS5 wind turbines when the yaw of the HS4 was equal to either $120°$, $180°$ or $240°$ (black line at $9D$). The orange arc highlights those range gates where the rotor of the HS1, HS2 and HS5 wind turbines are expected to be when the yaw direction of the HS4 is within the sectors $100° - 140°$, $160° - 200°$ and $220° - 260°$.

## 2.2 Post-processing of wind lidar data

Prior to the lidar data analysis, a quality check was performed in order to minimize biases that could originate from erroneous radial speed measurements. We present the post-processing of the two lidar data sets separately in the following two subsections.

### 2.2.1 *Wind Iris* wind lidar

In the analysis of the *Wind Iris* measurements, only the data flagged with a valid radial wind speed estimation was used. This data was selected based on the *Radial Wind Speed Status index* (*RWS Status index*) that is provided by the *Wind Iris* for each measurement (for more information we refer to the *Wind Iris* user manual (Avent Lidar Technology)). However, it was observed that on 84% of the 1-h periods examined, on average 0.2% erroneous estimations of the radial wind speed still remained after the valid data selection. Due to this, an additional filtering was applied, in which values outside the lower and upper outer statistical fences of the time series were treated as outliers for each 1-h period. The fences were defined by the first (lower fence) or third (upper fence) quartiles minus (lower fence) or plus (upper fence) 1.5 times the interquartile range of



the distribution of the radial wind speeds per period per range gate. The highest signal-to-noise ratio (SNR) values in all four line-of-sights were found approximately at 200 m, corresponding to the range where the *Wind Iris* has the highest sensitivity

and thereafter it decreased linearly (in dB) with distance. The corresponding data availability (defined here as the ratio between the selected and total acquired data) varied between 90% and 70%, with a decreasing trend for increasing measuring distance. In addition, the two lower beams presented a slightly lower availability compared to the two upper, which could be attributed to blade blockage.

### 2.2.2 *G4000 Galion* wind lidar

In the case of the *G4000 Galion*, the data filtering was based on the *intensity* values provided by the *Galion* software (Sgur-rEnergy Ltd., 2017), according to which the lower threshold value of 1.01 is recommended as the lower acceptable threshold for a valid measurement. This value corresponds to a SNR of $-20$ dB. Even though this is a conservative threshold, in relation to the ones typically used in profiling (Gryning and Floors, 2019) and scanning (Menke et al., 2020) wind lidars, an adequate data availability was observed in the acquired data set. In addition, in order to exclude erroneous wind estimation that were

observed mainly in the closest range gate an additional filter was constructed following the same method as the one applied to the *Wind Iris* data.

## 3 Methods

The parameterization of the upwind and wake conditions was performed using a right-handed coordinate system whose $x$-axis was anti-parallel to the axis normal to the rotor plane and thus aligned to the mean downwind direction in cases with no

yaw misalignment. Furthermore, we denote the three-dimensional wind vector $\boldsymbol{U} = \{u, v, w\} = \{u_1, u_2, u_3\}$, where $u_1$ is the component aligned with the $x$-direction. Using Reynolds decomposition, we express each of the three components as the sum of a mean (denoted with an over-bar symbol) and a fluctuating (denoted with a prime symbol) component: $u_i = \overline{u_i} + u_i'$.

### 3.1 Upwind radial speed model

The parameterization of the inflow wind conditions was performed by assuming that:

– the two components of the horizontal mean wind speed $\{\overline{u}, \overline{v}\}$, at a given height, are horizontally homogeneous within each measuring distance

– the vertical wind component is negligible ($\overline{w} = 0$ ms$^{-1}$)

– the wind shear $\frac{\partial \overline{u}}{\partial z}$ and veer $\frac{\partial \overline{v}}{\partial z}$ within the vertical range of the measuring area of the nacelle lidar are constant with height.

The first two aforementioned are justified by the horizontal homogeneity of the offshore mean wind field. The third assumes that in the vertical range between 68 m – 138 m (where the *Wind Iris* measurements are acquired) non-linearities in the wind





profile are weak in the measurement height. Based on the above assumptions, the mean wind vector in any given location with coordinates $(x, y, z)$ can be expressed only as a function of height as:

$$\boldsymbol{U}(z) = (\overline{u}, \overline{v}, \overline{w}) = \left( \overline{u}_\infty + \frac{\partial \overline{u}}{\partial z} \cdot z, \overline{v}_\infty + \frac{\partial \overline{v}}{\partial z} \cdot z, 0 \right) \tag{1}$$

where $\overline{u}_\infty$ and $\overline{v}_\infty$ are the mean free horizontal velocities at the height of the *Wind Iris* (103.4 m) and $\dfrac{\partial \overline{u}}{\partial z}$ and $\dfrac{\partial \overline{v}}{\partial z}$ are the vertical gradients. Further, the wind component $u$ is not expected to be homogeneous along the streamwise direction due to the induction by the wind turbine. We express this blockage effect using the induction factor $a$ that characterizes the reduction of the longitudinal wind speed along a line normal to the center of the rotor, based on the vortex sheet theory (Conway, 1995; Medici et al., 2011). This can adequately describe the wind speed evolution as it approaches a wind turbine rotor, as

demonstrated in a field test by Simley et al. (2016) in the case of an onshore wind turbine, using the following formula:

$$\frac{\overline{u}}{\overline{u}_\infty} = 1 - a \left( 1 + \frac{\zeta}{\sqrt{1+\zeta^2}} \right), \tag{2}$$

where $a$ is the induction factor which depends on the wind turbine operation and $\zeta = -x/R$ is the distance normalized by the rotor radius $R = 77$ m. Here, it is also assumed that the induction effect does not induce a spatial variability of the transverse component of the wind and that the induction is independent of $y$ and $z$ for the positions of the range gates. These simplifying

approximations are judged reasonable compared to the full expressions of Conway (1995).

Based on the above considerations the wind vector of Eq. 1 can be rewritten as:

$$\boldsymbol{U}(x, z) = \{ (\overline{u}_\infty + \frac{\partial \overline{u}}{\partial z} \cdot z) f_{\mathrm{ind}}(x), \overline{v}_\infty + \frac{\partial \overline{v}}{\partial z} \cdot z, 0 \}, \tag{3}$$

where

$$f_{\mathrm{ind}}(x) = 1 - a \left( 1 + \frac{-(x - x_L)/R}{\sqrt{1 + (-(x - x_L)/R)^2}} \right) \tag{4}$$

describes the expected reduction of the longitudinal speed across the whole rotor area, and $x_L$ is the distance between the rotor plane and the nacelle lidar (i.e. 4 m). Such parameterization formulations of the radial speed of a wind lidar have also been used in other studies of nacelle-mounted wind lidars (e.g. Borraccino et al., 2017).

The *Wind Iris* is acquiring radial wind speed $v_r$ measurements in four different upwind directions, which correspond to the projection of the wind vector $\boldsymbol{U}$ onto the unit vector of each of the line-of-sights of the wind lidar:

$v_r^{(i)} = -\boldsymbol{n}^{(i)} \cdot \boldsymbol{U}(x, z), \tag{5}$

where $\boldsymbol{n}^{(i)} = \{ n_1^{(i)}, n_2^{(i)}, n_3^{(i)} \}$ is the three-dimensional unit vector and the superscripts $i = 1, 2, 3,$ and 4 denote the index of each beam. The unit vector $\boldsymbol{n}$, which is defined by both the azimuth $\phi$ and elevation $\theta$ angles of the line-of-sight, and by the pitch $\phi_y$ and roll $\phi_x$ angles of the wind turbine's nacelle,

$\boldsymbol{n}^{(i)} = R_y(\phi_y) R_x(\phi_x) \boldsymbol{n}'^{(i)}, \tag{6}$





where $\boldsymbol{n}'^{(i)}$ is equal to:

$$
\begin{pmatrix} \boldsymbol{n}'^{(1)} \\ \boldsymbol{n}'^{(2)} \\ \boldsymbol{n}'^{(3)} \\ \boldsymbol{n}'^{(4)} \end{pmatrix} = \frac{1}{1+\tan^2\phi+\tan^2\theta} \begin{pmatrix} -1 & -\tan(\phi) & \tan(\theta) \\ -1 & \tan(\phi) & \tan(\theta) \\ -1 & -\tan(\phi) & -\tan(\theta) \\ -1 & \tan(\phi) & -\tan(\theta) \end{pmatrix},
\tag{7}
$$

where $\phi = \pm 15°$ and $\theta = \pm 5°$ depending on the line-of-sight direction.

Equation 5 can be written as:

$$
\begin{pmatrix} v_r^{(1)} \\ v_r^{(2)} \\ v_r^{(3)} \\ v_r^{(4)} \end{pmatrix} = -\boldsymbol{M} \begin{pmatrix} \overline{u}_\infty \\ \dfrac{\partial \overline{u}}{\partial z} \\ \overline{v}_\infty \\ \dfrac{\partial \overline{v}}{\partial z} \end{pmatrix} \quad \text{where} \quad \boldsymbol{M} = \begin{pmatrix} & & \vdots & & \\ n_1^{(i)} f_{\mathrm{ind}}(n_1^{(i)} d_f) & n_1^{(i)} n_3^{(i)} d_f f_{\mathrm{ind}}(n_1^{(i)} d_f) & n_2^{(i)} & n_2^{(i)} n_3^{(i)} d_f \\ & & \vdots & & \end{pmatrix}
\tag{8}
$$

is the matrix of the line-of-sight unit vectors and $d_f$ is the distance from the instrument to the measurement volume. In the
*Wind Iris* data, for each beam, an upwind distance $x_f$ is reported, which is the nominal $x$ distance from the instrument. Thus,
$d_f$ can be computed by multiplying $x_f$ by the norm of the vector $\{1, \tan\phi, \tan\theta\}$.

The estimation of the upwind parameters $\overline{u}_\infty, \overline{v}_\infty, \dfrac{\partial \overline{u}}{\partial z}, \dfrac{\partial \overline{v}}{\partial z}$ and of the induction factor $a$ was performed by solving Eq. 8.
The solution was found using a non-linear model solver in *Mathematica* (Wolfram Research), based on the 10-minute mean

values of the radial speeds in all available range gates. The solver had as input the initial estimation of the wind conditions
using only those radial wind speed measurements acquired in the furthest available range gate per 10-minute period. In this
analysis, we have selected only the 10-minute periods with an availability larger than $50\%$. The performance of the model, in
estimating the upwind mean wind speed characteristics, is investigated through the calculation of the root-mean-square error
(hereafter denoted as $\varepsilon_u$) between the modelled and measured radial wind speeds, for each 10-minute period.

## 3.2   Wake radial speed model

The study of the wake characteristics was performed by expressing the radial wind speed as a function of both the upwind
conditions (mean wind speed and direction) and the corresponding wind speed deficit that propagates downwind from the
wind turbine, using the following expression:

$$
v_r(x,y) = U\cos(\phi-\gamma)\left(1 - \frac{\beta(x)}{\sqrt{2\pi}\sigma(x)} \cdot \exp\left(-\frac{1}{2}\left(\frac{y-y_o(x)}{\sigma(x)}\right)^2\right)\right),
\tag{9}
$$

where $v_r$ is the radial wind speed measurement acquired by the scanning wind lidar in different longitudinal $x$ and transverse
$y$ distances in relation to the center of the rotor, $U = \sqrt{\overline{u}_\infty + \overline{v}_\infty}$ the mean speed of the horizontal wind at the height of the
wind lidar, $\phi$ the azimuth angle of the line-of-sight of the scanning wind lidar equal to $\arctan\dfrac{y}{x}$, $\gamma$ the mean wind direction
relative to the yaw of the wind turbine equal to $\arctan\dfrac{\overline{v}_\infty}{\overline{u}_\infty}$, $\sigma(x)$ a parameter representing the spread of the wake, $y_o(x)$ the

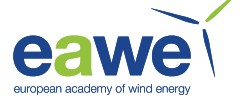


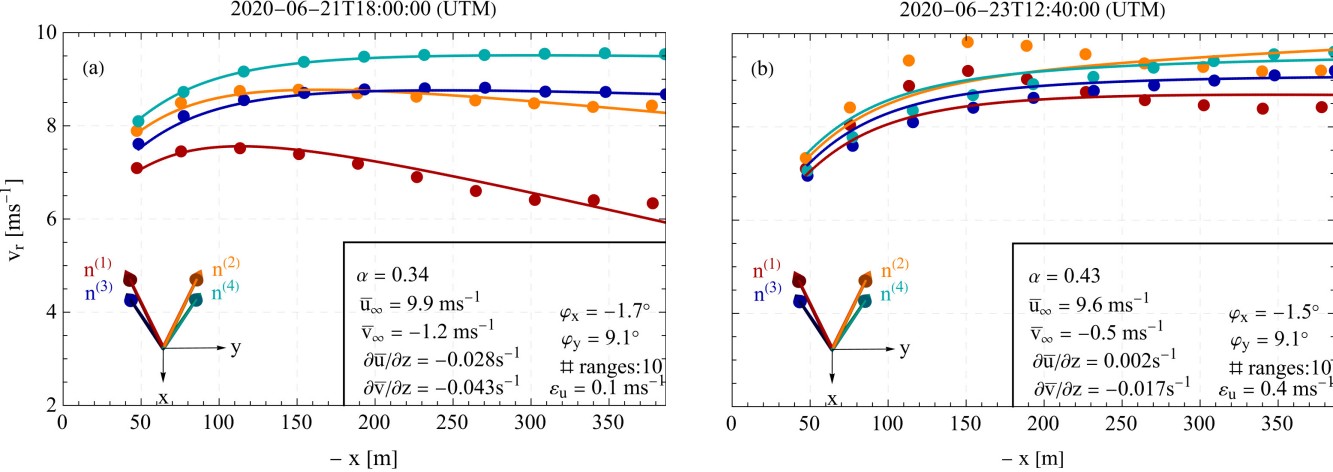

**Figure 4.** Example of two 10-minute periods where Eq. 5 describes either adequately (a) or poorly (b) the measured radial wind speed per upwind distance. The points correspond to the 10-minute mean radial wind speed measured by the *Wind Iris*, and the solid lines represent the corresponding simulated radial wind speeds of each of the four line-of-sights for the upwind distances between 50 and 400 m using the fitted parameters presented in the bottom right part of each plot. The negative $x$ axis denotes upwind distances in accordance with the coordinate system shown in Fig. 3.

center of the wake, defined by the location of the maximum wind speed deficit in the transverse direction at any downwind

distance, $y$ the transverse coordinate relative to the center of the rotor and $\beta(x)$ a scaling parameter representing the wind speed deficit. Using Eq. 9, we assume that the transverse profile of the flow in the wake can be described by a wind speed deficit with a Gaussian distribution. A similar wake model is presented in Aitken et al. (2014) and applied to nacelle-mounted wind lidar data in Aitken and Lundquist (2014). The parameterization of the radial speed measurements in the wake using Eq. 9 does not only enable investigating the propagation of the wake center due to wake meandering, but also the impact of

potential wind direction changes in the area scanned by the wind lidar, since each of the two aforementioned parameters induce a different pattern in the distribution of the radial wind speeds along the transverse axis. Similar to the analysis of the upwind conditions, the solution of Eq. 9 was computed using a non-linear model solver in *Mathematica* (Wolfram Research), based on the 30-minute mean values of the radial speeds. The solver was applied in each streamwise distance, and as an input of the initial estimation of the free wind and wake parameters, those measurements at $x/D = 6$ were chosen. The reason for selecting

that distance is that we had a sufficient number of measurements distributed in the transverse direction and a well-defined wake profile (an example is presented in Figure 10(c)).





# 4 Results

## 4.1 Free wind

During the approximately 3.5 month period when the *Wind Iris* was operating, 10529 10-minute periods of data were acquired
in total, corresponding to a 70% system availability. Equation 8, which describes the impact of the mean upwind conditions on
the measured radial wind speeds, was applied to 4502 10-minute periods, which were selected on the basis that, i. both *Wind
Iris* and SCADA data was available (9348), ii. the wind turbine was operating (7900), iii. radial wind speed measurements
were available in all ten ranges (7315), and iv. the yaw direction was between either $90°$ and $270°$ or $12.5°$ and $37.5°$ (4502)
to avoid cases where the *Wind Iris* measured partially or fully in the wake of one of the adjacent wind turbines. We selected
those wind directions based on both the wind farm layout (see Fig. 1(b)) but also by investigating the standard deviation of
the line-of-sight measurements of the two lower beams of the upwind-staring lidar. In that wind direction sector, we observed
consistent values of the standard deviation of the radial speeds. This finding is typical in free wind conditions, as shown by
Held and Mann (2019). The model's performance in estimating the free wind mean wind speed is investigated through the
calculation of the root-mean-square error (hereafter denoted as $\varepsilon_u$) between the modeled and radial wind speeds during every
10 minutes. Figure 4 presents two samples of 10-minute mean radial wind speed measurements of each of the four line-of-sight
beams (denoted in the figure as $\boldsymbol{n}^{(1)}$ and $\boldsymbol{n}^{(2)}$ for the two top beams and $\boldsymbol{n}^{(3)}$, and $\boldsymbol{n}^{(4)}$ for the two bottom beams and shown
as arrows), along with the simulated radial wind speeds using the estimated values of the upwind conditions and Eq. 8. The
figure presents two cases where the applied model is found to describe both accurately the wind conditions with relatively
low values of $\varepsilon_u$ (Fig. 4(a)) and inaccurately with relatively high values of $\varepsilon_u$ (Fig. 4(b)). In both plots, we observe that the
radial speed values of all beams decrease as wind approaches the wind turbine rotor. This is attributed to the induction of the
operating wind turbine. However, in Fig. 4(a), we see that the radial wind speeds of the two top beams are lower than those of
the corresponding bottom beams and have a decreasing trend as the upwind distance increases. This pattern corresponds to a
negative wind shear as it is predicted by the model in the case, where $\dfrac{\partial \overline{u}}{\partial z}$ is found to be equal to $-0.028$ s$^{-1}$. The reason why
the two right beams ($\boldsymbol{n}^{(2)}$ and $\boldsymbol{n}^{(4)}$) report higher values than the left ones ($\boldsymbol{n}^{(1)}$ and $\boldsymbol{n}^{(3)}$) is attributed to the negative value
of the mean transverse component of the free wind $\overline{v}_\infty$. On the other hand, in Fig. 4(b), even though we see a similar trend in
the two bottom beams with decreasing speed closer to the rotor, the two top beams present an increase in radial wind speeds
with upwind distance up to 180 m, followed by a decrease. The derived wind characteristics in both cases could be attributed
to a wind speed profile with an inversion. Such wind profiles are typically observed in either very stable atmospheric stability
conditions with a low atmospheric boundary layer height or in the case of low-level jets. Low-level jets are not uncommon over
offshore areas in Northern Europe. They are characterized by a non-linear wind shear (Emeis, 2014; Kalverla et al., 2019),
resulting in a high root-mean-square error $\varepsilon_u$.

To further investigate the relationship between the fitted values of the wind shear and veer and the resulting root-mean-square
error, we plot the fitted values of $\dfrac{\partial \overline{u}}{\partial z}$ and $\dfrac{\partial \overline{v}}{\partial z}$ as a function of the mean horizontal wind speed and the corresponding value
of $\varepsilon_u$ in Fig. 5. Negative wind shear cases are usually have $\varepsilon_u$ higher than 0.2 ms$^{-1}$. When $\varepsilon_u$ is lower than the value above,
the wind shear is usually within $0 - 0.02$ s$^{-1}$, which can be considered as low values, typical of offshore conditions. However,





an increasing trend of the wind shear values is found when the mean free wind speed is higher than 15 ms$^{-1}$. It is worth mentioning that among the estimated wind shear values with $\varepsilon_u < 0.02$ s$^{-1}$, 31% are negative. This shows that wind profile inversions are common at those heights (100 m – 150 m) and should be considered when studying the wind turbine operation. In the case of $\dfrac{\partial \overline{v}}{\partial z}$, we find generally negative values, which are expected from the Ekman spiral in the Northern hemisphere.

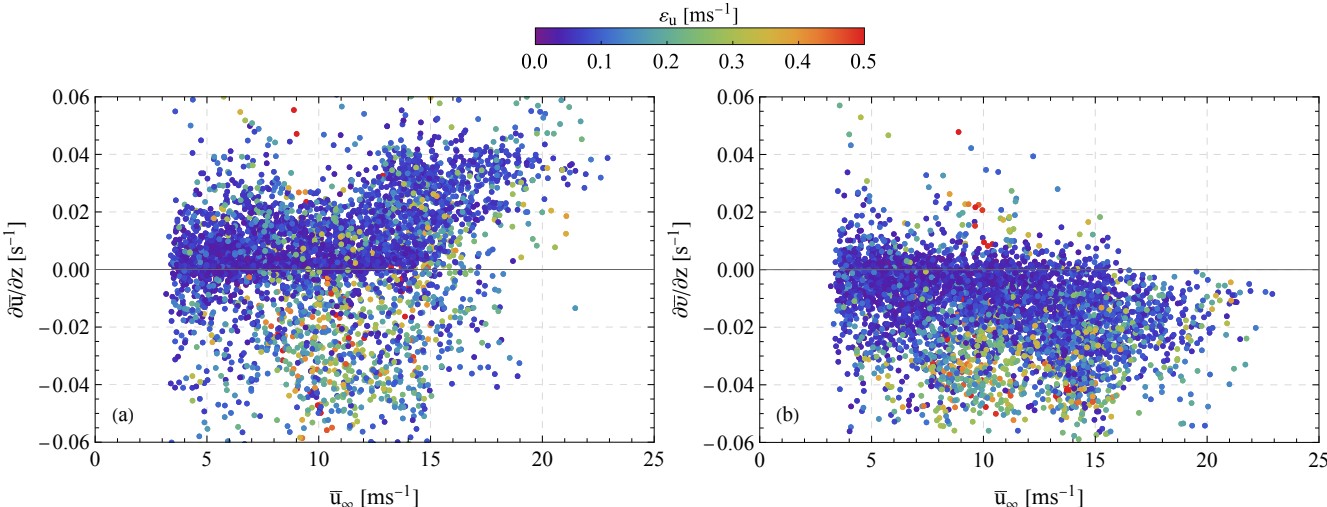

**Figure 5.** Scatter plot of the 10-minute mean wind sheer (a) and veer (b) versus the longitudinal component of the mean free wind speed. The color scale depicts the magnitude of the root-mean-square-error $\varepsilon_u$ of the fit of the 10-minute mean radial wind speed measurements of the *Wind Iris* to Eq.8.

For assessing the accuracy of the model of Eq.8, we compare the 10-minute mean horizontal wind speed at the hub height with the corresponding values of the nacelle-mounted anemometer, recorded in the SCADA system (see Fig. 6). For this purpose we use only cases where $\varepsilon_u < 0.2$ ms$^{-1}$. We find an overall good agreement between the two data sets, with a bias of less than 0.1 ms$^{-1}$ and a high Pearson correlation coefficient r>0.99. When studying the mean differences between the two data sets we find that the higher errors are observed between $7 - 9$ ms$^{-1}$ and between $18 - 22$ ms$^{-1}$. This error could be
attributed to a non-optimal transfer function used in the nacelle-mounted anemometer.

Furthermore, the *Wind Iris* data was used to estimate the turbulence intensity of the ambient wind conditions. For this purpose, the measurements acquired closest to the rotor distance (i.e., 50 m) were used. This selection was based on i. having the smallest spatial separation of the two measuring points that can introduce a bias in the estimated second order statistics (i.e., standard deviation of $u$), ii. the vertical displacement between the two line-of-sights due to the roll angle was on average small,
since a $1°$ mean roll angle induced an average vertical displacement of less than half a meter. Moreover, the estimated statistics is not expected to be biased by the wind turbine operation, as it was demonstrated by Mann et al. (2018), where they have showed the induction zone influences mainly the low-frequency fluctuations. Furthermore, at those range gates, we observe the maximum data availability. The estimation was performed by using the 0.25 Hz time series of the radial wind speeds, by solving the equation $v_r^{(i)} = \{n_1^{(i)}, n_2^{(i)}\} \cdot \{u, v\}$. In the calculation of the standard deviation of $u$ we don't take into account the





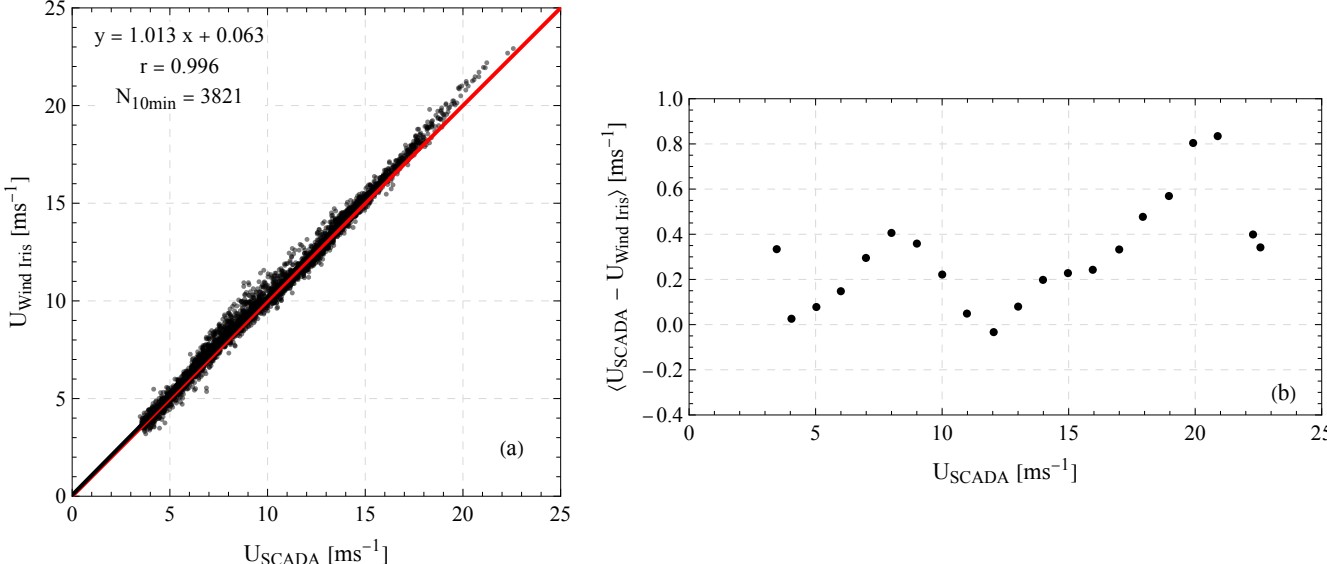

**Figure 6.** (a) Scatter plots of the estimated free wind speed $U_{\text{Wind Iris}}$ at hub height using the *Wind Iris* measurements and the corresponding 10-minute mean value $U_{\text{SCADA}}$ as measured by the nacelle-mounted anemometer. (b) Bin averaged mean difference $U_{\text{SCADA}}$ - $U_{\text{Wind Iris}}$ as a function of $U_{\text{SCADA}}$.

contribution of the nacelle's motion to the radial speed. Thus, the estimated TI values are expected to be slightly biased, as it is presented in the work of Gräfe et al. (2022).

## 4.2 Induction zone

In Fig. 7, the induction factor $a$ is presented as a function of the estimated free horizontal wind at the height of the *Wind Iris*. The color of each point denotes the corresponding rmse $\varepsilon_u$ values of Eq. 8. An increase in the $\varepsilon_u$ values results in a spread of 290 the estimated induction factor values per wind speed. In general, we observe that the induction factor is on average equal to $0.37 \pm 0.07$ in the below rated wind speed range, between 4 ms$^{-1}$ and 10 ms$^{-1}$. These values are close to the theoretical Betz limit, assuming that the operation of a wind turbine can be simulated using an actuator disk model. Above that wind speed range, the induction factor is decreased until it reaches $0.05 \pm 0.02$ when the wind speed gets larger than 18 ms$^{-1}$. Overall, we do not observe any correlation between the induction factor and the estimated wind speed shear or veer, which would indicate 295 that the wind turbine, for a given wind speed, responds differently when the shear or veer changes.

## 4.3 Wake

In addition to the selection criteria listed in Section 4.1 for the characterization of the upwind flow, four additional criteria were needed for the wake study. First, only *G4000 Galion* wind lidar data acquired during periods when the mean tilt angle of the

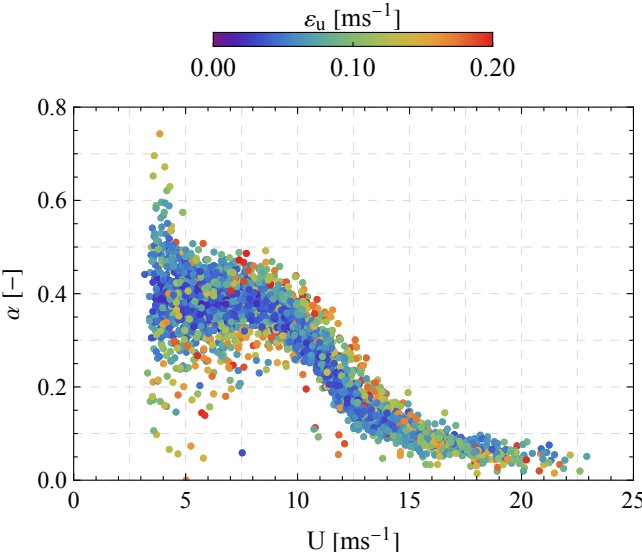

**Figure 7.** Estimated values of the induction factor $a$ per 10-minute mean horizontal wind speed using Eq. 4. The colors denote the corresponding rmse $\varepsilon_u$ values.

wind turbine was within $4.425°$ and $5.575°$ were selected to ensure that the measurements of the PPI scan were approximately

horizontal. This tilt angle range results in a vertical displacement of the furthest range gate by $\pm D/10$, equivalent to half the

dimension of one grid cell. Further, to avoid systematic biases introduced by spatial variations of the data availability, only

those scans with a data availability of 75% or higher were selected. 5907 scanning pattern iterations remained after applying

the two criteria above. Subsequently, to study the wake characteristics in relation to the upwind conditions, we first had to

identify those 10-minute periods where: i. both SCADA and *Wind Iris* data were available (1719), ii. the wind turbine was

operating (1719), iii. the inflow was free from wakes (1121), and iv. the measured radial wind speeds of the *Wind Iris* could be

modelled using Eq. 8 with relatively low errors defined by a $\varepsilon_u < 0.2 \, \mathrm{ms}^{-1}$ (868).

Figure 8 presents four examples of wake measurements acquired over different 2-minute periods. The examples correspond

to cases with below, and above-rated mean wind speeds and two turbulence intensity levels equal to 2% and 4%. We observe

that regardless of the turbulence intensity in the above-rated speed (Fig. 8(b) and (d)), the trace of the wake is visible down to

9 rotor diameters, without a visible wake expansion. In the case of an above-rated wind speed, when the turbulence intensity

is equal to 2%, we observe a very low standard deviation of the roll angle ($\sigma_{\phi_x}$). Jacobsen and Godvik (2021) studied the

response of the wind turbines at *Hywind Scotland* and observed that very low standard deviations of the roll angle were usually

associated with stable atmospheric stratification. In the current example, the wind shear appears to be negative, concordant with

a very stable atmosphere, which could explain the propagation of the wake down to 9 rotor diameters. However, we observe

that the wake deficit is still visible but less strong when both the TI and the wind shear increase. Below rated, the 2-minute

scans showed less systematic characteristics. In the top left plot (Fig. 8(a)), we present a case with a negative shear and low



**Figure 8.** Four examples of radial wind speed measurements in the wake flow acquired during individual PPI scans over a 2-minute period. The examples correspond to cases with two different levels of turbulence intensity of $\sim \%2$ (top row) and $\sim \%4$ (bottom row), for mean wind speeds below (left column) and above (right column) rated speed. The boxes above each plot list the upwind conditions, as well as the Strouhal number of the side-to-side motion, the standard deviation of the roll angle and the yaw direction of the wind turbine.





standard deviation of the roll angle, indicating a stable atmospheric stratification. The wake deficit is visible down to 4 rotor diameters. Furthermore, a reduced wind speed deficit in the range between 1 and 2 rotor diameters downstream of the nacelle is visible. However, after 4 rotor diameters, the wake is less evident. Instead, we observe wavy patterns in the distribution of the

radial speeds that are spread in the longitudinal and transverse directions. This feature may be artificially created by the flow characteristics and the relative slow scanning speed of the wind lidar. The PPI scan starts from the positive $y$-axis and rotates towards the negative, possibly explaining the direction of the "stripes" in the plot. This feature is not a necessary characteristic of all the below-rated wind speed cases, as shown on the bottom left plot, where the wake trace reaches $9D$ and meandering is visible. In the plot, we can also see the effect of the induction zone of one of the adjacent wind turbines (i.e. HS2).

The selected data were gathered in 30-minute periods and averaged to produce 170 cases of a time-averaged wake in a fixed frame of reference. Furthermore, in the analysis, we chose only periods characterized by a stationary time series of the longitudinal wind speed, based on the assumption that the time series are quasi-stationary when their dependence on time has a slope of less 0.5 ms$^{-1}$h$^{-1}$. The quasi-stationary periods were identified by applying a linear regression of the 0.25 Hz time series of the longitudinal component measurements from the upwind wind lidar. Finally, we selected only those 30-minute

periods that have at least 7 PPI scans, which resulted in 89 cases, in which the mean free wind speed was between 8.2 and 14.8 ms$^{-1}$ and the TI varied between 1.4% and 8.1% (see Fig. 9). For those cases, we found that the fluctuations of the tilt angle of the wind turbine were either correlated or anti-correlated to the time series of the longitudinal wind speed depending on whether the mean wind speed was below or above rated speed (i.e., 10 ms$^{-1}$), respectively. The mean peak-to-peak difference of the tilt angle, for the selected cases equals $1.8° \pm 1.1°$.

On the contrary, the roll angle of the nacelle was characterized by a stronger periodic sinusoidal motion with a varying phase and a mean peak-to-peak difference equal to $0.9° \pm 0.6°$ (see also the power spectral densities presented in Fig. B1). This side-to-side motion corresponds to a mean lateral displacement of the rotor equal to approximately 98.5 m $\times \tan \dfrac{0.9}{2} = 0.7$ m or less than $1\% D$. The small variations of both the pitch and the roll angle show that the wind turbine is relatively stable regardless of the wind conditions, in accordance with the findings reported by Jacobsen and Godvik (2021), where neither the

mean wind speed nor the atmospheric stability had a significant impact on the wind turbine response.

In our analysis, we used the variations of the pitch angle to estimate the standard deviation of the velocity of the rotor and the frequency $f_x$ of the periodic side-to-side motion to estimate the Stouhal number $St$ for each of the selected cases as $\dfrac{f_x D}{\overline{u}_\infty}$. We find that the standard deviation of the longitudinal velocity of the rotor is, on average, six times smaller than the standard deviation of the wind speed. Therefore, we can assume that in this field study, the impact of the longitudinal motion of the

wind turbine's rotor on the wake conditions is negligible compared to the effect of the random variations of the wind speed. Furthermore, given the side-to-side motion's low amplitude, we studied the wake characteristics based on the mean wind speed and turbulence intensity values. In Fig.9, we present the number of cases with the same mean wind speed and turbulence intensity, using a 1-ms$^{-1}$ and 2% range resolution, respectively.

In Fig. 10, we present the mean and the standard deviation, over a 30-minute period between 12:30 – 13:00 on 01/09/2020,

of the radial wind speeds in the downwind side of the wind turbine. By averaging all scans of the period, we minimize the effect of both random noise and biases due to the natural fluctuations of the wind speed and direction during a single scan

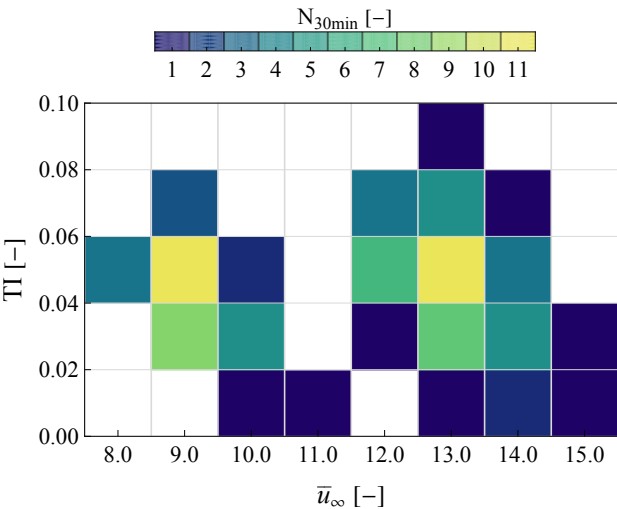

**Figure 9.** Two dimensional array of the selected 30-minute periods as a function of the mean speed and turbulence intensity of the free wind.

(e.g. Fig 8(a)). Furthermore, we should expect that the time-averaged wake has a smaller deficit and wider distribution that an instantaneous snapshot of the wake flow. The measured plane covers the downwind propagation of the wake, where wind speed deficits can be traced down to 1500 m (9.7 $D$). The wake is concentrated within an area with approximately the same width as

the wind turbine rotor. The standard deviation of the radial wind speed reveals two bands of high turbulence at the edges of the wake, which correspond to the areas where the wake mixes with the free flow.

An example of the application of the wake model of Eq. 9 is presented in Fig. 10 (c). Applying a Gaussian model to fit the wake deficit is not always suitable for describing the characteristics of the flow. We assess the model's suitability by investigating the root-mean-square error $\varepsilon_d$ values between the model described by Eq. 9 and the radial speed profile per

downwind distance for each of the 89 cases selected for this study. In Fig. 11, we present the mean values of $\varepsilon_d$ per downwind distance. At close distances, $\varepsilon_d$ takes values above $0.4\,\mathrm{ms}^{-1}$, but decreases with distance. This trend is attributed to the fact that at short distances, the measurements are distributed over a limited transverse range of the PPI scan, and that the profile of the wake cannot typically be described by a single Gaussian deficit (e.g. Aitken et al., 2014). We define the acceptable threshold of valid application of the model as $\varepsilon_d < 0.25\,\mathrm{ms}^{-1}$. Using this criterion, we conclude that the adequate model area is within the

range $2.7 < x/D < 8.3$. The increase in $\varepsilon_d$, at the range $8.3 < x/D$, is attributed to both the lower SNR of the measurements at those distances, which typically results in higher random errors, and to the fact that occasionally the flow in those ranges was further distorted by the induction zone of the adjacent wind turbines.

To investigate the accuracy of Eq. 9, we assumed that the free wind speed is constant along the whole measured area where wakes are absent. Then, the ensemble average of the free wind speed $\langle \overline{U} \rangle$ is calculated based on the mean of the free horizontal

wind values, $U$, estimated in each downwind distance between 2.7 and 8.3 rotor diameters. Subsequently, the estimations are compared to the ones derived using the *Wind Iris* data. The regression analysis was performed using a least squares perpendicular distance method (Deming fit). Except for a few outliers, we find that the down- and upstream estimations have a reasonable



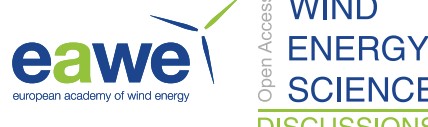

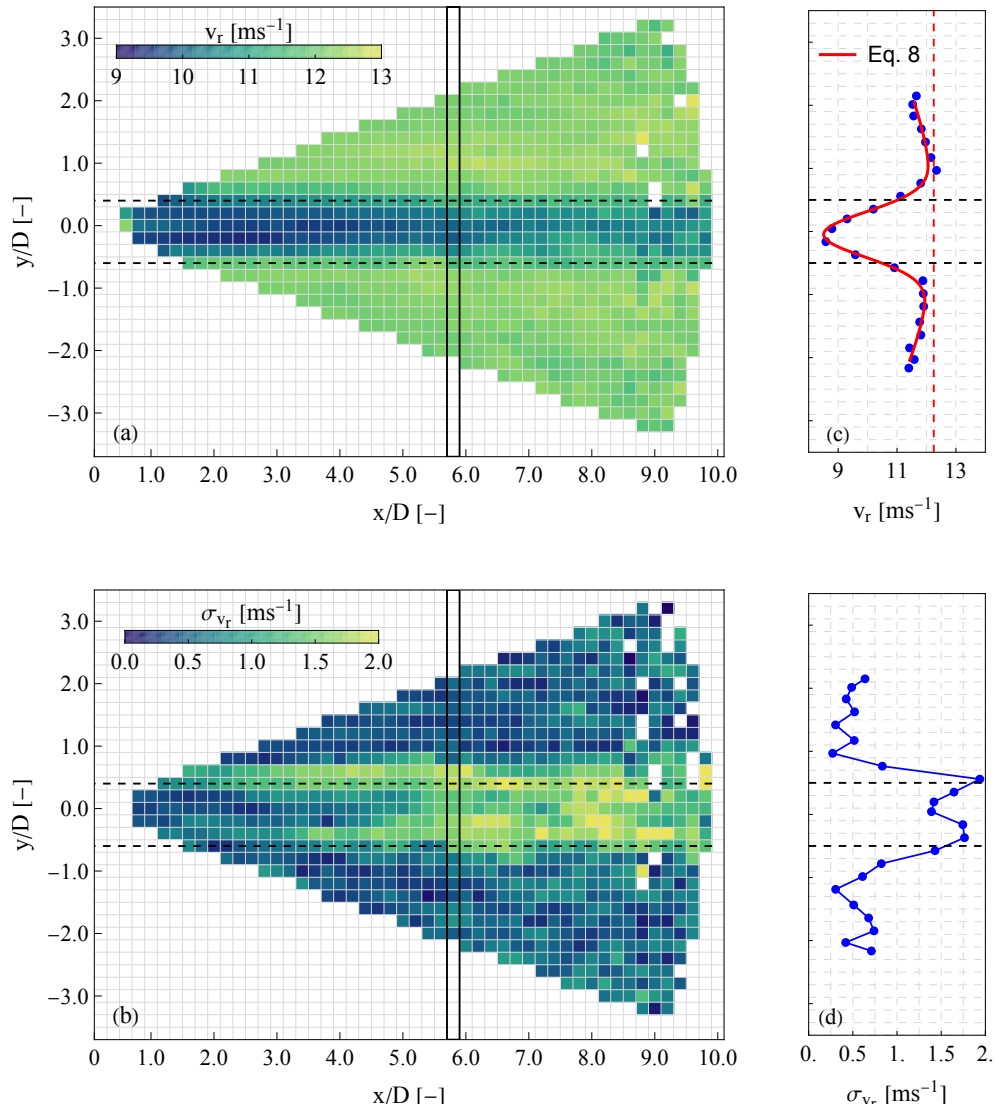

**Figure 10.** Mean (a) and standard deviation (b) of the radial speed of a 30-minute period over the scanned area. An example of the transverse profile of the mean (c) and standard deviation (d) of the radial speed at $x/D = 6$. The result of the application of Eq. 9 is presented in red in plot (c).

correlation and a slight bias ($\sim -0.1$ ms$^{-1}$, see Fig. 12(a)). We also find that the mean yaw misalignment is equal to $0.1° \pm 1.8°$ based on the upwind staring nacelle lidar, while from the downwind scanning lidar it is $-0.2° \pm 2.8°$ (see Fig. 12(b)).





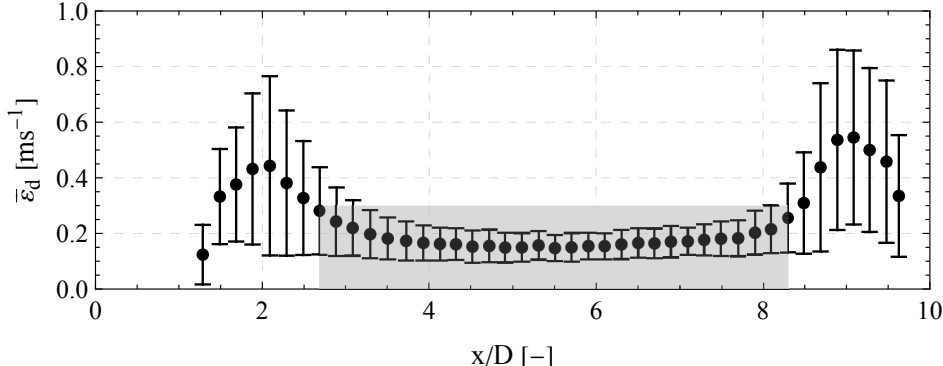

**Figure 11.** The mean (black dots) and standard deviation (error bars) of the root-mean-square error $\varepsilon_d$ between the acquired radial wind speeds and the Gaussian wake deficit of Eq.9 for different downwind distances. The gray opaque rectangular area denotes the limits $2.7 < x/D < 8.3$ over which, on average, we observe acceptable $\varepsilon_d$ values and thus denotes the range over which the wake characteristics are resolved.

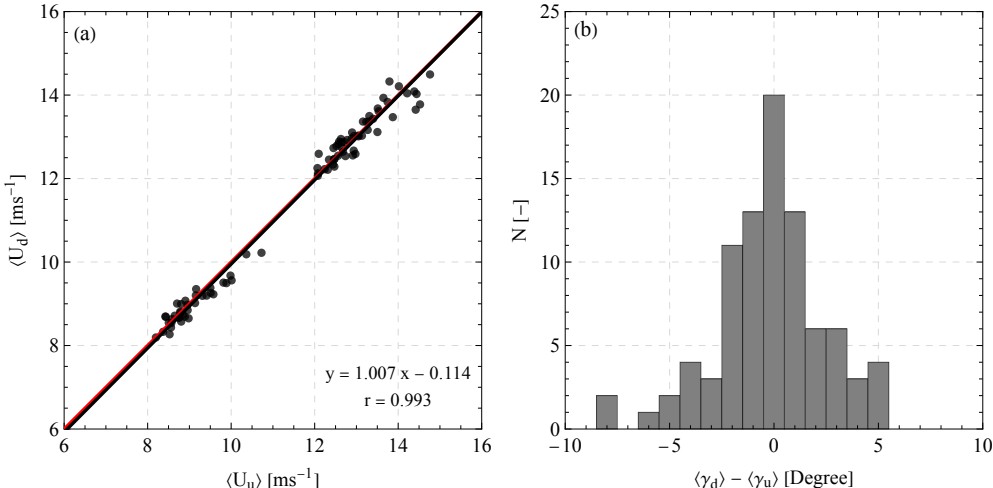

**Figure 12.** Scatter plot of the 30-min free wind speed (a) and histogram of the mean yaw misalignment (b) estimations from the *Wind Iris* ($\gamma_u$) and the *Galion* ($\gamma_d$) wind lidars. The black and red lines in (a) represent the results of the regression analysis (black) and the identity (red) line. The regression analysis results and the corresponding correlation coefficients are presented on the bottom right of the plot.

### 4.3.1 Self-Similarity in the far-wake

To investigate the similarity in the far-wake, we calculate the *self-similar velocity defect* $f$ as a function of the *scaled cross-stream variable* $\xi$, which, according to Pope (2000, Ch. 5), are defined as:

$$f(\xi) = \frac{\overline{u}_\infty - \overline{u}(x,y)}{\overline{u}_\infty - \overline{u}(x,y_0)} \quad \text{and} \quad \xi = \frac{y}{y_{1/2}}, \tag{10}$$





where $y_{1/2} = \sqrt{2\ln 2}\sigma$ and $\sigma$ correspond to the half-width of the half maximum of the wind speed deficit, and the standard deviation of the Gaussian distribution of the wind deficit, $\overline{u}_\infty$ is the mean free wind speed, $\overline{u}(x,y)$ the mean longitudinal wind speed, and $y_0$ the center of the wake. For estimating the $\overline{u}$ values in the wake, we first assume that the mean vertical speed component is zero. This assumption is based on the offshore wind conditions and on the *far-wake* region of the wake, being mainly related to the ambient wind conditions (Vermeer et al., 2003). Thus, one should not expect any rotational motion in the wake flow after $x/D > 3$, as seen in Zhang et al. (2012). Therefore a mean radial wind speed $\overline{v_r}$ measurement at a given point $\{x,y\}$ is equal to the projection of the $u$ and $v$ components to the line-of-sight direction:

$$\overline{v_r}(x,y) = \overline{u}(x,y)\cos(\phi - \gamma) + \overline{v}(x,y)\sin(\phi - \gamma), \tag{11}$$

where $\gamma$ and $\phi$ denote the yaw misalignment and the azimuth direction of a light-of-sight, respectively.

Since the $\phi$ angles are small $\{-20°, 20°\}$ and the mean absolute yaw misalignment, derived from the estimated values of $\overline{u}_\infty$ and $\overline{v}_\infty$, is equal to $\gamma = 1.5° \pm 1.1°$, we can neglect the term $\overline{v}\sin(\phi - \gamma)$ without introducing a bias. Thus, through Eq. 11 we can express the mean $u$ component as:

$$\overline{u}(x,y) = \frac{\overline{v_r}(x,y)}{\cos(\phi - \gamma)}. \tag{12}$$

Thus, the longitudinal wind speed at the center of the wake $y_0$, using Eq. 9 and 12, is equal to:

$$\overline{u}(x,y_0) = U\left(1 - \frac{\beta(x)}{\sqrt{2\pi}\sigma(x)}\right). \tag{13}$$

The results using Eq.12 and Eq.13 in Eq.10 are presented in Fig. 13 for a 30-minute period with a mean free speed equal to 13 ms$^{-1}$. We find that the distribution of the velocity profile of the wake along the lateral axis to the wake propagation can be considered self-similar for distances between $2.2 < x/D < 9.6$. We determine this visually, since the estimated profiles of the velocity defect $f$ can be expressed as a function of $\xi$ (?, Ch. 5) Furthermore, we find that the velocity defect profiles are in a good agreement with a Gaussian distribution, defined as $e^{-\ln 2\xi^2}$. This observation supports the hypothesis of a uniform turbulent viscosity in the far-wake flow of this case study (van der Laan et al., 2023).

Subsequently, we apply the same analysis for all 89 30-minute selected periods. Between $3 - 10$ rotor diameters, we find relatively low rmse $\varepsilon_f$ values between the estimated and modelled velocity deficit profile. This result supports the hypothesis that the profile of the velocity deficit is generally self-similar and, thus, independent of the downwind distance, regardless of the mean free wind speed and the corresponding turbulence intensity. The results here agree with the findings of Chamorro and Porté-Agel (2009), who, in a wind tunnel study, found that the vertical profile of the wind speed deficit in the wake developed both over a smooth and rough surface is approximately symmetric.

## 4.4 Case studies

Before studying the wake characteristics for different ambient free wind conditions, we investigate the variability in the wake measurements between 30-minute periods with the same mean wind speed and TI values. In Fig. 15, we present 11 profiles of the mean longitudinal wind speed at the center of the wake when the 30-minute mean wind speed is equal to 9 ms$^{-1}$ and TI



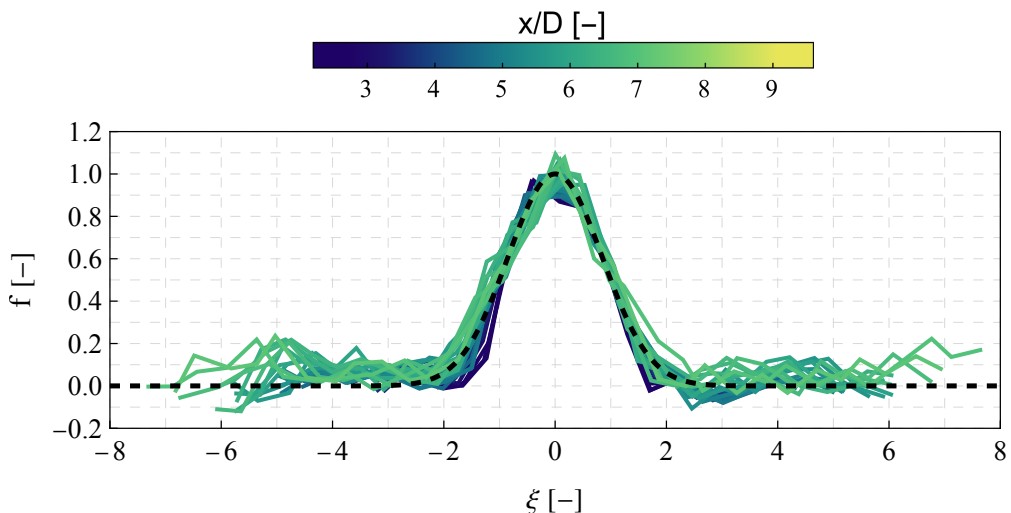

**Figure 13.** Mean profile over a 30-minute period of the self-similar velocity defect $f$ as a function of the *scaled cross-stream variable* $\xi$ for different downwind distance between $2.2 < x/D < 9.6$. The dashed line corresponds to the theoretical distribution $e^{-\ln 2\xi^2}$ of an axisymmetric wake.

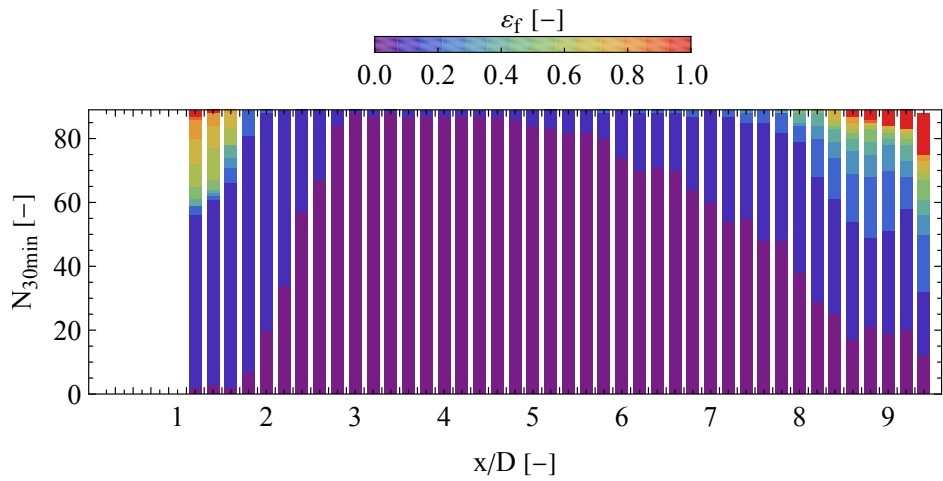

**Figure 14.** Bar charts of the estimated root-mean-square error $\varepsilon_f$ of applying a Gaussian function (Eq. 10) to the spanwise radial velocity measurements for different downwind distances between 1 and 10 rotor diameters.

equals to 5.0%. The wind speeds were normalized by dividing with the mean horizontal wind speed of each 30-minute period estimated using Eq. 9. The profiles are generally in agreement in the region $x/D < 8$. Case #1, where a faster wake recovery (smaller velocity deficit as the downwind distance increases) is observed, is an exception, with no apparent correlation with the wind shear or veer values. Furthermore, in profiles #2, #5, #9, and #10, we see a decrease in the radial wind speeds when





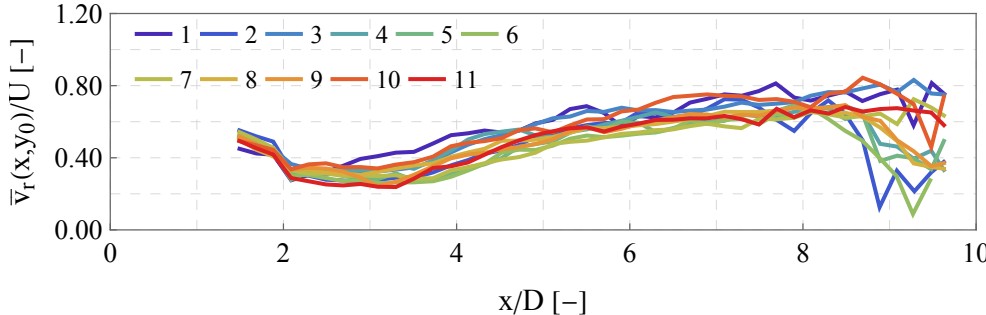

**Figure 15.** Wind profiles of the velocity in the wake center normalized by the mean wind speed $\overline{u}_{\infty,d}$ for 11 cases when the 30-minute mean wind speed was equal to 9 ms$^{-1}$ and the corresponding TI was equal to 5%.

**Table 1.** The number of 30-minute periods $N_{30min}$ and the corresponding characteristics of the upwind profile (longitudinal speed $\overline{u}_\infty$, turbulence intensity TI, yaw misalignment $\gamma$, shear $\frac{\partial \overline{u}}{\partial z}$ and veer $\frac{\partial \overline{v}}{\partial z}$, $\sigma_u$) at the hub height, the wind turbine's induction zone ($a$), the standard deviation of the wind rotor speed ($\sigma_{v_{\text{rotor}}}$) and the Stouhal number of the side-to-side motion ($S_t$) for each of the five cases examined.

| Case | $N_{30min}$ [#] | $\overline{u}_\infty$ [ms$^{-1}$] | TI [%] | $\gamma$ [°] | $\frac{\partial \overline{u}}{\partial z}$ [s$^{-1}$] | $\frac{\partial \overline{v}}{\partial z}$ [s$^{-1}$] | $a$ [-] | $\sigma_{v_{\text{rotor}}}$ [ms$^{-1}$] | $S_t$ [-] |
|------|------|------|------|------|------|------|------|------|------|
| I | 5 | 9.0±0.3 | 3.4±0.5 | -1.0±0.6 | 0.008±0.007 | -0.009±0.003 | 0.38±0.01 | 0.05±0.01 | 0.09±0.07 |
| II | 12 | 8.8±0.2 | 5.0±0.7 | -0.0±2.0 | 0.010±0.007 | -0.009±0.008 | 0.38±0.04 | 0.07±0.02 | 0.19±0.17 |
| III | 7 | 13.1±0.2 | 3.3±0.4 | 0.8±1.3 | 0.023±0.010 | -0.026±0.006 | 0.15±0.02 | 0.09±0.06 | 0.10±0.15 |
| IV | 11 | 12.9±0.3 | 4.7±0.5 | 1.4±0.7 | 0.013±0.011 | -0.010±0.010 | 0.15±0.01 | 0.10±0.02 | 0.10±0.10 |
| V | 5 | 12.7±0.2 | 6.4±0.6 | 0.4±0.9 | 0.011±0.008 | -0.011±0.006 | 0.15±0.02 | 0.11±0.04 | 0.02±0.01 |

$x/D$ is between 8.5 and 10. This decrease is attributed to the presence of the induction zone of neighbouring wind turbines.

Therefore, we apply an additional filter by remove measurements beyond 8.3 rotor diameters. Moreover, those 30-minutes periods with negative shear are also excluded from the analysis, in order to avoid cases with inversions in the upwind wind profile.

After applying the two additional filtering criteria, we examine two cases where the mean wind speed is either below (i.e. 9 ms$^{-1}$) or above (i.e. 13 ms$^{-1}$) rated speed. In those two mean wind speed values, we have the maximum number of cases

based on Fig.9 and thus enables the estimation of the statistical variability of the derived mean parameters. The data from those cases are split and examined based on their TI values. The overall wind conditions of those cases and the corresponding upwind profiles are presented in Table 1 and Fig. 16, respectively. We observe that, on average, the wind shear and veer have similar values for all cases. An exception is case III, where a stronger shear and veer are found. In all cases, the absolute yaw misalignment is less than 2° and we find a Stouhal number ($St$) of the side-to-side motion between 0.1 and 0.2, besides the

case V where low values of $St$ are observed.



**Figure 16.** Vertical profiles of the mean longitudinal $\overline{u}$ and transverse $\overline{v}$ components of the wind vector for each of the five cases examined. The measurement height is normalized by the hub height $h$, and the two horizontal dashed lines in each plot denote the lower and upper limits of the wind turbine's rotor. The error bars correspond to the standard error of each estimated mean value.





### 4.5 Wake characteristics

For each of the selected five cases we study the wake characteristics in terms of the downwind propagation of the wake center's position, the wake's spread, and the maximum velocity deficit.

#### 4.5.1 Wake center

The propagation of the wake center $y_0$, normalized by the rotor diameter $D$, for each of the five cases is presented in Fig. 17 . In cases I and II, Fig. 17(a), we observe that the wake center is translated towards the negative $y$-axis with an increasing trend with the downwind distance down to approximately $7D$. The observed translation of the wake center could be partially attributed to the estimated yaw misalignment angles $\gamma$ presented in Table 1. A yaw misalignment of $2°$ could result in a transverse displacement by 0.3 at $8D$. On the contrary, in cases III – V, even though the estimated yaw misalignment angles are of the

same magnitude, the mean wake center, presented in Fig. 17(b), shows hardly any translation. Specifically, it is within $0.05\ D$ and $-0.10D$ from the center of the rotor within the downwind range $3-8D$, with, however, larger standard error $\sigma_{y_0}/N_{30\text{min}}$ values (denoted as error bars in the figure). We could not find any correlation between these trends and any other free wind condition characteristics. In all cases, the standard error of the wake center increases with the downwind distance, which could be attributed to the meandering of the wake.

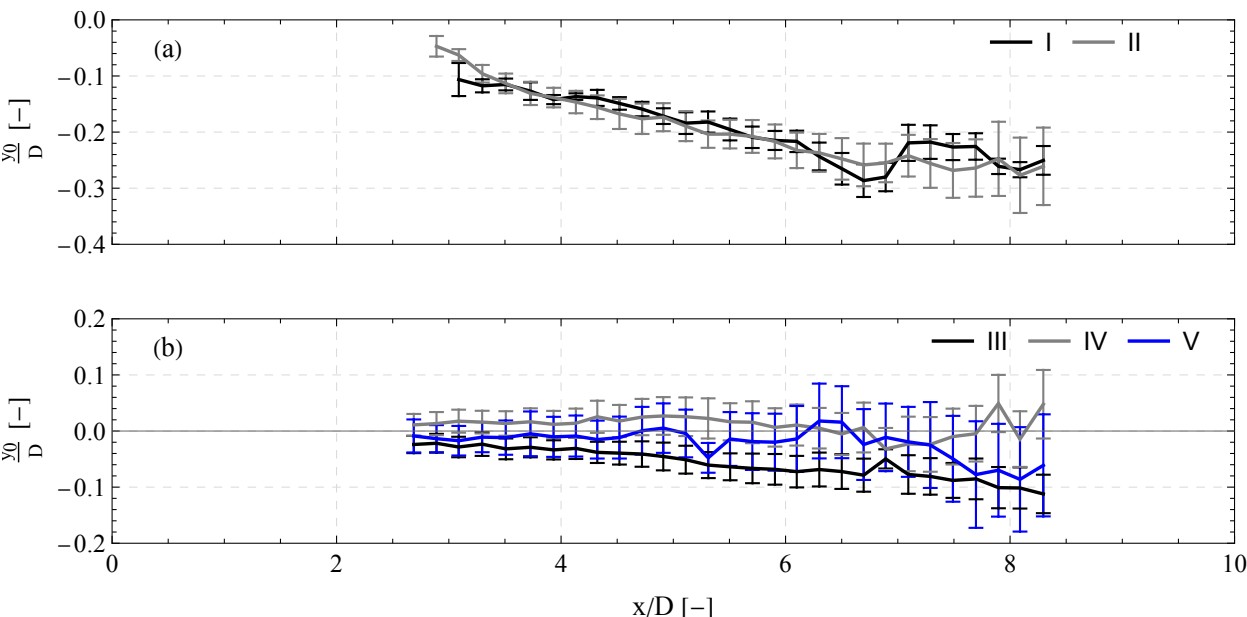

**Figure 17.** Mean wake center along the downwind distance between 2.7 and 8.3 rotor diameters in a fixed frame of reference, when the mean wind speed is equal to (a) $9\ \text{ms}^{-1}$ (below rated speed) and (b) $13\ \text{ms}^{-1}$ for different TI levels that range from 3.4 % to 6.3 % (see Table 1). The error bars correspond to the standard error of each mean value.





## 4.6 Wake width

Similarly to Aitken et al. (2014), we define the wake width equal to $4\sigma$, i.e., the spanwise range where the $95\%$ wind speed deficit is concentrated. The values are normalized by the rotor diameter $D$ and presented in Fig. 18. In all cases, we observe an increase in the wake width with downwind distance. Below rated (Cases I & II, Fig. 18(a)), already at a downwind distance of three rotor diameters, the wake width is approximately equal to $1.5 - 2.0D$ and increases up to 2 times at eight rotor diameters. Furthermore, as the ambient TI increases from 3.4 to 5.0% the wake width also increases. Above rated (Cases III – V) in Fig. 18(b), we find a slowly growing width as a function of downwind distance, with values ranging between $1.3D$ and $2.0D$. The relatively constant values indicate a slow wake recovery in the TI range 3.3 % – 6.4 %.

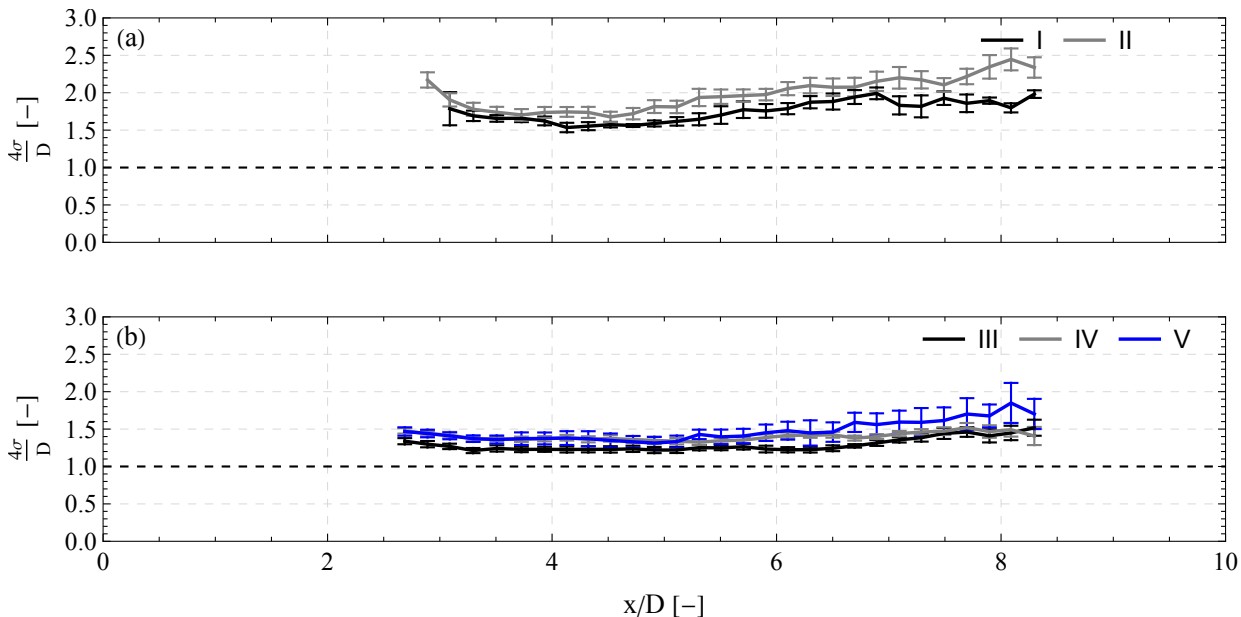

**Figure 18.** Mean wake width along the downwind distance between 2.7 and 8 rotor diameters in a fixed frame of reference when the mean wind speed is equal to (a) $9\,\mathrm{ms}^{-1}$ (below rated speed) and (b) $13\,\mathrm{ms}^{-1}$ for different TI levels. The error bars correspond to the standard error of each mean value.





### 4.7 Velocity deficit

The velocity deficit in the center of the wake can be expressed as the relative difference between the mean longitudinal wind component in the center of the wake along the spanwise axis $\overline{u}(x,y_0)$, and the free wind speed $\overline{u}_\infty$. Thus, using Eq. 13, the velocity deficit in the wake center is equal to:

$$\frac{\overline{u}_\infty - \overline{u}(x,y_o)}{\overline{u}_\infty} = \frac{\beta(x)}{\sqrt{2\pi}\sigma(x)}, \tag{14}$$

where the $\beta(x)$ and $\sigma(x)$ correspond to the fitted values in Eq. 9. In Fig. 19, we present the percentage of the velocity deficit for different downwind distances extending from 2.7 to $8.3D$, when the mean free wind speed is 9 ms$^{-1}$ (Cases I — II) and 13 ms$^{-1}$ (Cases III – V), respectively. Below rated, in Fig. 19(a), we observe a velocity deficit around $69\% - 78\%$ at $3D$ that decreases to $38\% - 41\%$ at $8D$. The estimated percentages of the velocity deficit have relatively low standard errors that depending on the downwind distance range between $0.7 - 3.1$. In the below rated wind speed case, we observe that a $1.6\%$ increase in TI results in a velocity deficit decrease of $7.5\% \pm 3.0\%$ in the range between 3 and 8 rotor diameters, which is independent from the downwind distance. The observed high velocity deficit values at $x = 3D$ agree with the high induction factors presented in Fig. 7. By taking into account the induction factor and Eq. 2, we should expect a maximum relative velocity deficit equal to $2a\times100\%= 76\%$ (which is presented by a dashed black line in Fig.19(a)). We observe that the velocity deficit at $3D$ is approaching the theoretical maximum relative velocity deficit. However, in the case above rated, in Fig. 19(b), we observe that the velocity deficit spans from around $37\% - 38\%$ at 3 rotor diameters and decreases with increasing distance down to $27\% - 29\%$ at 8.3 rotor diameters. The observed velocity deficit at $3D$ is almost 10% higher than the theoretical based on the estimated induction factor ($a$=0.15). Furthermore, an increase of the TI from 3.3% to 4.7% and from 4.7% to 6.4% results to a decrease of the velocity deficit of $5.7\% \pm 3.4\%$ and $9.9\% \pm 3.8\%$, respectively. This decrease is becomes larger as the downwind distance increases. Overall, the observed decrease in wind speed deficit is a lot slower than the cases I and II. Specifically, we find that wind deficit scales with the downwind distance on the power of -0.4, -0.5, and -0.6 for cases III – V, respectively. These values are lower than in cases I and II where -0.9 and -0.9 is found, which is in close agreement with the findings of Barthelmie et al. (2004). In all cases, III, IV, and V, the wind speed deficit between $2.3D$ and $3.5D$ is higher than the expected magnitude based on the corresponding estimated induction factor values.

## 5 Discussion

To investigate the impact of the ambient atmospheric wind conditions on the wake flow generated by floating wind turbines, we assumed that: i. the wind shear and veer are constant throughout the wind turbine rotor area, ii. the wake flow, on overage, propagates horizontally, iii. rather than the atmospheric stability, turbulence plays the dominant role in the wake recovery and iv. estimations of turbulence intensity can be derived by combining radial speed measurements acquired at a 50 m upwind distance and a spanwise separation of 26.8 m ($0.17D$).

Regarding the first assumption, a constant and height-independent wind sheer and veer was used to parameterize the upwind conditions and simulate the radial wind speed measurements of the *Wind Iris* lidar. In 74% of the cases, we have found a good





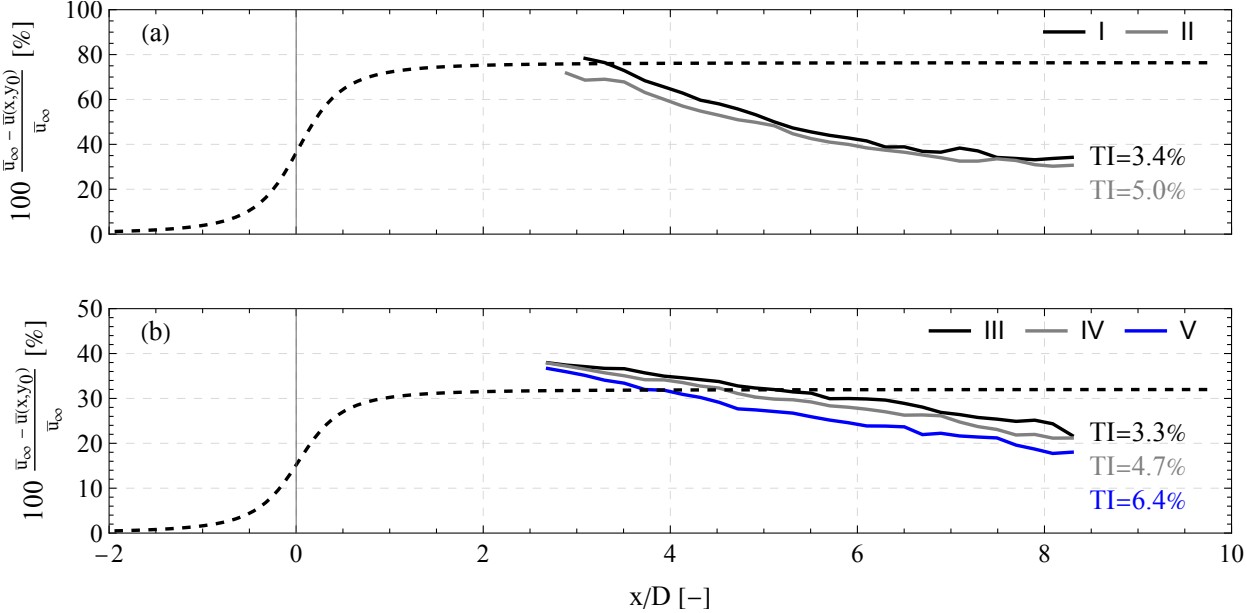

**Figure 19.** Mean velocity deficit at the wake canter along the downwind distance between 2.7 and 8 rotor diameters, when the mean wind speed is equal either to 9 ms$^{-1}$ (a) or 13 ms$^{-1}$ (b) below rated speed) and for different TI level. The gray dashed line in each plot represents the predicted longitudinal speed using Eq. 2 and the induction factor values of $a=0.38$ (a) and $a=0.15$ (b).

agreement between the *Wind Iris* measurements and the upwind model of Equation 3. Cases with a non-constant wind sheer and veer can be attributed to low atmospheric boundary layer height or low-level jets. In either case, measurements at higher and lower heights would provide insight into a more comprehensive description of the vertical profile and are recommended in future studies.

     Furthermore, since we used only PPI scans in this analysis, it was not possible to study if vertical motions of the wake

occurred, which could introduce a bias in the observed velocity deficits. For this reason, we had to hypothesize that the mean wake propagation takes place horizontally. Observations that support this hypothesis have been acquired by the *Galion* wind lidar on HS2 that was measuring in a RHI scanning mode (see more in Appendix A). The measurements of that lidar could not be analyzed in depth since, due to the layout of the wind farm (see Fig. 1), and for the wind direction sector examined in this study (southern winds), the inflow of HS2 included wake flows from adjacent wind turbines. However, when we investigate

the measurements from that lidar for Northern wind directions and for the same wind speed ranges (see Fig. A1), we observed that on average we could not detect a significant vertical motion of the center of the wake (see Fig. A2). Overall, assessing the wind characteristics with a scanning lidar is challenging, but also offers a lot of possibilities. The acquired data sets point to the direction that wind lidars are technologically mature to be used in commercial applications in offshore conditions. As a future best practice, we recommend a combination of PPI and RHI scanning configurations to study the mean characteristics

of floating wind turbine wakes.



A limitation of this study was that the lack of information about atmospheric stratification. This knowledge would allow a more thorough investigation of the wake properties and their dependence on atmospheric conditions. However, the relatively high speeds, the strong shear values at the examined heights and the low standard deviation of the roll angles (based on the findings of Jacobsen and Godvik (2021)) support the hypothesis that the data set was acquired during either neutral or stable
atmospheric stability and thus, the impact of convective effects of the wake is limited.

Finally, regarding the impact of the motion of the floater on the wake characteristics, the amplitude of the surge and sway motions used in this study was relatively small compared to the values used in numerical (Nanos et al., 2021; Chen et al., 2022) and wind tunnel (Fu et al., 2019; Schliffke et al., 2020) studies that focused on the study of the effect of the wind turbine motion on the wake characteristics. Comparing the standard deviation of the surge motion speed with the longitudinal speed showed
that the standard deviation of the wind was one order of magnitude larger. For this reason, we studied the wake characteristics as a function of the mean wind speed and TI values. Similar amplitudes of side-to-side motion with the one observed in our study, with, however, an almost double Strouhal number, have been reported in the work of Li et al. (2022). They state that the side-to-side motion amplifies the wake meandering when the ambient TI is low ($< 3\%$), leading to a faster recovery of the wake deficit. However, in the TI regime studied here 4 % − 8 %, the wake recovery of a floating wind turbine is similar to
that of a fixed wind turbine, as reported by (Li et al., 2022). This similarity in wake recovery could explain the agreement of the wake characteristics presented in this analysis with results from previous numerical and wind tunnel studies of fixed wind turbine wakes.

## 6 Conclusions

This study uses two nacelle-mounted wind lidars to investigate the characteristics of the up- and downwind conditions relative
to an offshore floating 6MW wind turbine. The acquired radial wind speeds from the two lidars are parameterized as a function of the upwind and wake characteristics. In the case of the upwind conditions, we find that, in 74% of the cases, modelling the radial speeds as a function of a linear wind shear and veer along the rotor, and of an induction factor due to the wind turbine's operation, performed adequately. Over the examined data set, we found that in 31% of the cases, the wind shear had negative values. These negative values highlight that deviations from a linear increasing profile at a height range between
100 m and 150 m are not negligible in offshore wind profiles. The wake study focused on the downwind propagation of the wake's center, width and deficit in the far-wake region. The knowledge of these parameters is important since they determine the inflow conditions that the adjacent wind turbines would encounter in a wand farm. The wake measurements were grouped based on the mean wind speed at the hub height and the corresponding TI, allowing a statistical wake property analysis. Our findings support the hypothesis that the spanwise velocity profile in the wake can be considered self-similar in the far-wake
region (3 − 8 rotor diameters), as well as it can be modelled adequately using a constant eddy viscosity. Finally, an increase in the ambient TI enhance the recovery of the velocity deficit. These results indicate that the wakes of floating wind turbines, that do not experience high surge and sway motions, will have similar characteristics to those of fixed wind turbines. Thus,



even though the rotor of floating wind turbines is subject to motions induced by both wind and sea fluctuations, the primary mechanism for the wake recovery is the atmospheric turbulence intensity.





## Appendix A: Range height indicator lidar scans

This appendix presents the range height indicator (RHI) scans performed by the *G4000 Galion* wind lidar installed on the HS2 wind turbine. The scans were performed along a vertical plane centered in the nacelle. Figure A1 presents the mean longitudinal speed, estimated by dividing the radial wind speed measurements with the cosine of the elevation angle of the line-of-sight. The estimation of the mean is based on two different data sets based on the wind speed range of 7.4 – 10.5 ms$^{-1}$ and 10.6 – 13.5 ms$^{-1}$. The two sets had 426 and 266 scans for the first and second wind speed ranges, respectively. The data are grouped only based on the mean wind speed.

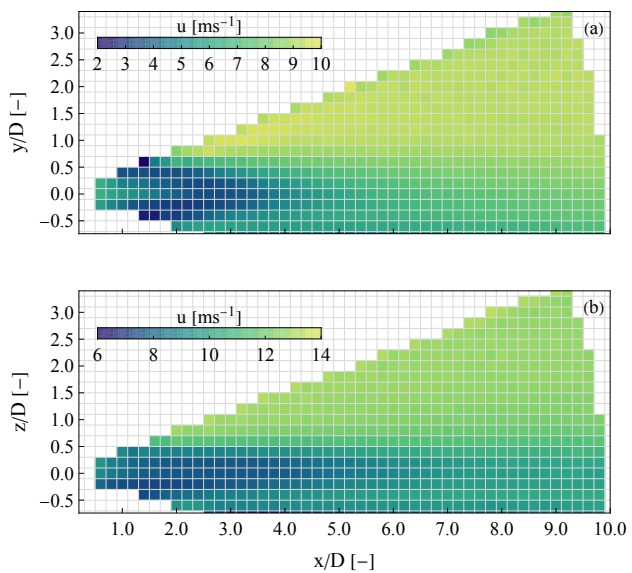

**Figure A1.** Mean longitudinal wind speed acquired during a range height indicator scan for the two different wind speed ranges of (a) 7.4 – 10.5 ms$^{-1}$ and (b) 10.6 – 13.5 ms$^{-1}$.

Figure A2 presents the vertical profile of the wind speed based on the RHI scans of Fig. A1 for six different downwind distances corresponding to 3, 4, 5, 6, 7, and 8 rotor diameters. Each profile is normalized by the mean free wind speed at the hub height. The profiles show a visible velocity deficit for all the ranges down to $8D$. Furthermore, the maximum velocity deficit is near the hub height regardless of the downwind distance. On average, the wake center shows no significant vertical motion even though the nacelle tilt is 5°.

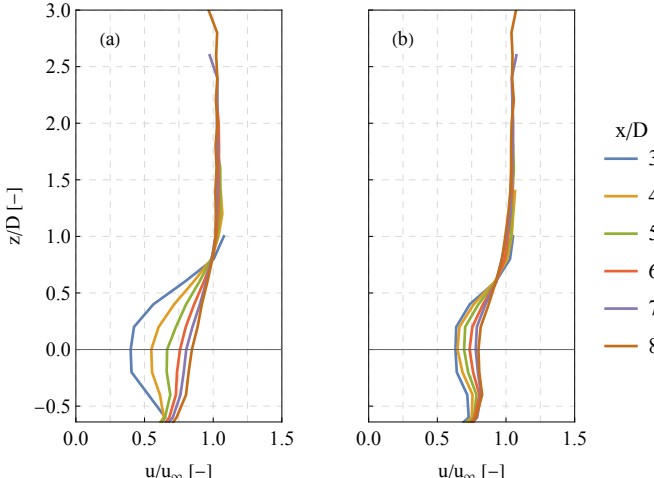

**Figure A2.** Vertical profiles of the longitudinal wind speed at different downwind distances (3, 4, 5, 6, 7, and 8 $D$) and for two wind speed ranges of (a) $7.4 - 10.5$ ms$^{-1}$ and (b) $10.6 - 13.5$ ms$^{-1}$.

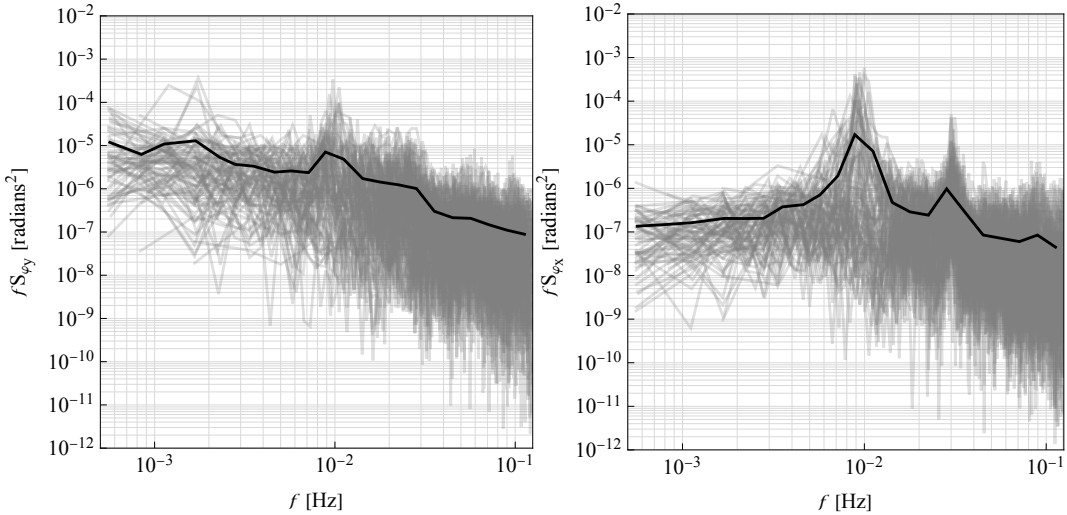

**Figure B1.** Power spectral densities of the pitch (left) and roll (right) angles of the wind turbine measured by a motion reference unit (MRU) in the nacelle. The gray lines correspond to the individual spectra for each of the 89 cases of Fig. 9, and the black lines depict the corresponding mean over all cases.



## Appendix B: Pitch and roll angle of the wind turbine's nacelle

*Author contributions.* Hywind Scotland wind farm and CDB planned the campaign and performed the measurements; NA and JM conceptualized the research analysis; NA analyzed the data and prepared the data visualization; NA wrote the original draft of the manuscript; NA, JM and CDB reviewed and edited the manuscript


*Competing interests.* At least one of the (co-)authors is a member of the editorial board of Wind Energy Science.

*Acknowledgements.* Hywind Scotland is acknowledged for providing access to the data. This work was in part supported by the Horizon Europe project FLOW (HORIZON-CL5-2021-D3-03-04, grant no. 101084205).





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
