# Peer review of "Revealing inflow and wake conditions of a 6MW floating turbine"

_Wind Energy Science, 2023_

## Referee Comment (RC1)

**Review of 'Revealing inflow and wake conditions of a 6 MW floating turbine'**

**by Angelou et al.**

**Summary of the article:**

The authors present novel results on the wind upwind and downwind flow characteristics of a floating offshore wind turbine using LiDARs. To my knowledge, this is the first time such an extensive measurement of the wake of a floating wind farm has been performed. In this study, the turbine wake is characterized (wake center, velocity deficit, upwind and crosswind radial wind speed gradients) in relation to a range of parameters (atmospheric stability, wind speed, turbulence intensity, distance downwind, LiDAR sampling statistics, yaw misalignment).

**Significance:**

This study makes a first and important methodological contribution to the field, by suggesting the use of LiDARs, that have the advantage of providing both temporal and spatial measurements of the wave. This study is especially interesting considering the great scarcity of field observations of floating turbine wakes. Some limitations to the study are highlighted.

**General overview:**

The research topic is well presented in the introduction, and the abstract clearly states the study objectives. The article presents a new methodology that is well adapted to the study of the wake of floating offshore wind turbines. This approach consists in the deployment of wind LiDARs. The scientific results are very promising. The results are produced through the analysis of a new and valuable dataset, as a result of a rigorous experimental setup.

Despite the very promising dataset, the article struggles to clearly transmit the key messages of the article to the reader. The article's structure is confusing amid mistakes, repetitions, and an unnecessary count of 22 figures, some of which are not presented or described in the text. Furthermore, the study does not comment and discuss the results at sufficient depths. For example, the authors seem to observe contradicting effects of the yaw on the center of the mean wake center (Sect. 4.4) and struggle to explain the seemingly absent correlation between shear, vear with the induction factor (Sect. 4.2) but do not sufficiently investigate possible explanations, and do not bring these issues to the discussion and conclusion. This is unfortunate, as these observations should be highlighted, so as to allow for improvement in future work.

In conclusion, I believe that this article should be published. However, major corrections are required in order to improve language, structure, and physical interpretations. I advise the authors to thoroughly read the article again to find possible typos, and provide a more concise and dynamic description of the work. Please find my comments and suggestions below.

**Major comments/questions :**

- I find that the article is too long, boasting 22 figures. In Section 4.3 you lose control of your figures, and begin to list them instead of integrating them into an organized presentation of your results. Furthermore, you do not (from what I have seen) talk about or present Fig. 10D. In the same fashion, I have not found where you present and discuss Figs. 11 and 12. If you don't discuss them, you may have to remove them. However, they are interesting. So you should maybe consider reducing the overall size of the article, and focus on the results that you do want to clearly present.

- What were the operating characteristics of the turbine ? What is the influence of operational variables such as turbine rotation speed, blade angles, etc, on the wake ? Maybe you don't have access to this data, but then you must very clearly state that you have not considered this, and that this is remaining research to be done on this (which should be indicated in the discussion and conclusion).

- In the discussion, I highly recommend that you make a clear summary of your assumptions, and the limits of your study. Among this, you should mention that you assume stationary conditions over your 30 minute periods, etc.

- What are the possible effects of ocean waves on the motion of such a platform ? Can we expect an interaction between the wake and atmospheric stability ? I think this should be better clarified in the conclusion, to provide an idea of what the perspectives are.

- In the conclusion, no strong link has been made between the fact that this is a 6 MW turbine. What is the significance of this study in the context of other wind turbines deployed around the world on floating platforms ?

- You should consider changing the structure of the article. Indeed, Section 4.5 'Wake characteristics' is very much the same as Section 4.3 'Wake'. Maybe 4.5 should be a subsection of 4.4, as these are the case studies.

- In section 4.5.1, you study the center of the wake. I think this deserves a deeper discussion considering that this is an active field of research, especially in the case of floating offshore turbines. Also, I am not convinced as to why, in Figure 17, cases I and IV are not the same. Is this because the higher winds are damping the transverse displacements ?

- You should check the whole text for clarity, and to correct the text in respect of editorial guidelines. For example, write 'Figure' and not 'Fig.' at the beginning of a sentence. When you present a figure for the first time, make a dedicated sentence to present it (axes, colors, curves, etc): 'Figure X presents...'. Other examples are listed below.

**Minor comments:**

L19: Reformulate sentence. What do you mean by a 'good wind resource' ?

L19-21: I suggest that you reformulate the sentence

L84: Is 1 Hz enough ? Why ?

L89: Do you know why there is a 3% decrease ?

L102: Please clarify 'area covered by the rotor'

L109: After the end of the sentence is probably the best place time to explain how long it takes to make a full PPI scan, as we don't want this important information to be buried later down in the text

L122: I wonder if this paragraph should be in this article. You only show this in the appendix. Also, is the HS2 the same as HS4 ?

Figure 3: The orange curve is not very clear. Maybe select a more visible color ?

Sect. 2.2.1: Can you clarify if you used the SNR to filter data ? Did you use a SNR threshold ?

Sect. 3.1: You present your assumptions. What about stationarity ? Do you assume that you are stationary in terms of incident wind speed conditions at 10 minutes ? At 2 hours ? I would appreciate that you mention stationary here, and later in the text.

L167: replace by 'weak at measurement height' and you should provide references for this assumption

L179-180: Maybe discuss what possible effects you are neglecting. How could this be affected by attitude (pitch etc) ? I see you already refer to Borraccion et al. 2017 later, maybe try to link the sentences ?

L258: Could you explain what you know of the stability on that day, so as to support your argument?

L264: You should explain the figure in the text. Also, please clarify this paragraph.

L266: Maybe you should show a fitting function to help show the trend. As it is hard to know where all your points are, it would be interesting to statistically illustrate your trend (add a fitting function, and provide statistical values).

L271: You are presenting Fig. 6 in brackets at the end of the sentence. Figure 6 deserves better, and should be presented and described in a clearer manner. You may also remove the mention of the SCADA, as this has already been presented before.

L280-282: This carries an important message. This deserves to be made more obvious, and move out of the middle of the paragraph.

L292: I suggest that you add references on actuator disk models.

L295: Please clarify sentence, as I do not understand it. From what I currently understand, it seems that the non-correlation is explained by the fact that they may be correlated.

L306: Maybe you should explain why TI is different for similar U10 wind speeds ?

L315: Please specify in what figure I can see this wind shear increase etc

L325: As of this paragraph, you use 30 minute, 2 minute, and 10 minute averages. Could you clarify this choice ? Maybe you can detail that you will be doing this a bit earlier, such as at the beginning of the section. It is quite hard to follow when you bring new averaging types at the beginning of each paragraph without prior notice.

L321-324: Could you provide some clearer explanation of why Fig. 8D does not show these 'stripes' ?

Figure 9: I wonder how useful this figure is to the discussion.

Figure 10: Maybe consider showing the slices (such as in Fig. 10c) for other x/D values ? I also realize that you do not refer to Fig. 10D in the text. If you don't mention it, then consider removing it. Finally, maybe you can remind what LiDAR you used in the caption of the figure.

Figures 11 and 12: They do not serve much in the discussion, maybe they should be removed ? However, I think that Figure 11 is quite interesting, and carries a more interesting and simpler message that Figure 14 that could be removed instead, and replaced with a couple of sentences.

Figure 13: I don't see the values at the higher end of the x/D range (near 9.6 in yellow), but they should appear on the front.

Figure 14: Also present the vertical axis

Figure 15: The label should specify 'horizontal velocity' ?

Figure 16: You show the constant du/dz and dv/dz, which you assume at the beginning of the article. Maybe you should make this clear, as this is quite a nice result.

L430: You should consider reminding the reader how you are detecting the wake center. Also make a proper presentation of Fig. 17.

L434: Where does this 2° value come from ?

L461: what is 'a' in the equation ?

**Typos etc:**

L8: Replace by 'along the horizontal plane' ?

L23: replace by 'realistically model', or '...flows realistically'

L31: Add 'For example, the results…'

L32: remove 'for example'

L32: remove 'to' by 'in', add comma after 'recovery'

L39: 'Focused'

L45: Replace 'enhancing' by 'increasing'

Figure 1 label: Replace 'whose y-axis' by 'where the y-axis'

L72: Remove 'have been'

L80: Replace 'relative to that' by 'of the'

L81: End sentence at nacelle. New sentence: 'The MRU measured the rotation…'

L86: Replace 'Section' by 'Sect.'

L87: New paragraph after 'longitudinal'. And the longitudinal what ? Longitudinal axis ?

L95: add 'used for this study were the Wind Iris' …

L101: Replace 'Leveled' by 'Level'

L103: Replace 'points' by 'dots' . Replace 'defining' by 'that define'

L109: replace 'spanned from' by 'spanning'

Sect. 2.2.2: Maybe add something like 'As described in Sect. 4.3, the Galion will be filtered for cases where the scans are horizontal'.

L153: Replace 'upwind and wake conditions' by 'upwind and downwind' ?

L171: Remove 'Further'

L172: A repetition, as you also define the induction factor alpha later (L177).

L191: Replace 'i=1,2,3 and 4' by 'i=1,2,3,4' ?

Sect. 3.2: Maybe it would be interesting to present why you want a radial speed model. What is you objective here ?

L221: I suggest a new paragraph after 'deficit', and write 'We assume, using Equation 9…'

L228: I suggest 'The solver was applied in each streamwise distance, and the measurements at x/D = 6 were chosen as an input…'

L229: Suggest 'This choice is supported by the sufficient number of measurements'

L234: Suggest 'a total of 10529' and remove 'in total' later

L236: End sentence at 'periods'. Begin new sentence with 'They were selected'

L241: I suggest a new paragraph here

Figure 4: Consider moving the figure to this page

L277: Suggest replacing 'selection' by 'choice'

L293: Replace 'gets larger than' by 'exceeds'

L318-319: Replace with 1D, 2D, 4D

L324: Provide reference to figure instead of 'In the plot'

L325: Provide reference to the 'selected data', which is this selected data ?

L394: You should properly introduce Fig. 13

L397: Error with references

---

## Author Comment (AC1)

**Answer to Referees**

Nikolas Angelou[1], Jakob Mann[1], and Camille Dubreuil-Boisclair[2]

[1]Technical University of Denmark, Department of Wind and Energy Systems, Frederiksborgvej 399, 4000, Roskilde, Denmark
[2]Equinor ASA, Sandslivegen 90, 5254, Sandsli, Norway

**Correspondence:** Nikolas Angelou (nang@dtu.dk)

**1 Answer to Referee 1**

**1.1 General overview**

1. The research topic is well presented in the introduction, and the abstract clearly states the study objectives. The article presents a new methodology that is well adapted to the study of the wake of floating offshore wind turbines. This approach consists in the deployment of wind LiDARs. The scientific results are very promising. The results are produced through the analysis of a new and valuable dataset, as a result of a rigorous experimental setup.

Despite the very promising dataset, the article struggles to clearly transmit the key messages of the article to the reader. The article's structure is confusing amid mistakes, repetitions, and an unnecessary count of 22 figures, some of which are not presented or described in the text. Furthermore, the study does not comment and discuss the results at sufficient depths. For example, the authors seem to observe contradicting effects of the yaw on the center of the mean wake center (Sect. 4.4) and struggle to explain the seemingly absent correlation between shear, vear with the induction factor (Sect. 4.2) but do not sufficiently investigate possible explanations, and do not bring these issues to the discussion and conclusion. This is unfortunate, as these observations should be highlighted, so as to allow for improvement in future work.

In conclusion, I believe that this article should be published. However, major corrections are required in order to improve language, structure, and physical interpretations. I advise the authors to thoroughly read the article again to find possible typos, and provide a more concise and dynamic description of the work. Please find my comments and suggestions below.

We would like to thank Referee #1 for the positive feedback and for the constructive criticism in the manuscript. Referee #1 provided a thorough review of our manuscript that included both major and minor comments. We have tried to answer all the questions and comments, as well as to implement the correction of the typos that the reviewer has highlighted.

Regarding: i. the length of the article we have reduced it by removing Figs. 9, 10(b) & (d), 12 and 14, ii. the comment on the induction factor (Sect. 4.2) please see our answer to the comment #19 in Sect. 1.3 and iii. the comment on the wake center please see our answer to the comment #31 in Sect 1.3, in our answer to the referees.

**1.2 Major comments/questions**

1. I find that the article is too long, boasting 22 figures. In Section 4.3 you lose control of your figures, and begin to list them instead of integrating them into an organized presentation of your results. Furthermore, you do not (from what I have seen) talk about or present Fig. 10D. In the same fashion, I have not found where you present and discuss Figs. 11 and 12. If you don't discuss them, you may have to remove them. However, they are interesting. So you should maybe consider reducing the overall size of the article, and focus on the results that you do want to clearly present.

   (a) In order to reduce the length of the article we removed Figs. 9, 10(b) & (d), 12 and 14 (The figure numbers correspond to the ones in the original version.

      – Figure 9 presented an array plot of the number of 30-minute periods in different mean wind speeds and turbulence intensity levels. The reason for including that figure was to highlight the challenge that we experienced in acquiring measurements in similar wind conditions and to justify the selection of the case studies presented in the article. In the revised version the figure is removed.

      – Figures 10(b) and (d) were implicitly and very briefly discussed in lines 355 and 356 of the original version of the manuscript. We have now removed them and instead, following a suggestion of the reviewer added two more downwind profiles of the mean wake.

      – Figures 12 and 14 were removed and the results presented in those plots are now mentioned in the text.

   (b) Figure 11 is discussed in lines 360 – 367 of the original version of the manuscript.

   (c) Figure12 is discussed in lines 370 – 374 of the original version of the manuscript.

   (d) In order to improve the comprehension of the Sect. 4.3, we have decided to split the original text of that section in two subsections entitled "Single wake scan" and " 30-min mean wake scan".

   (e) Furthermore, we restructured sections 4.3 to 4.7 of the original article as following:

      – 4.3 Wake
         – 4.3.1 Single wake scan
         – 4.3.2 30-min mean wake scan
         – 4.3.3 Self-similarity in the far-wake
         – 4.3.4 Case studies
         – 4.3.5 Wake center
         – 4.3.6 Wake deficit
         – 4.3.7 Velocity deficit

2. What were the operating characteristics of the turbine ? What is the influence of operational variables such as turbine rotation speed, blade angles, etc, on the wake ? Maybe you don't have access to this data, but then you must very clearly state that you have not considered this, and that this is remaining research to be done on this (which should be indicated in the discussion and conclusion).

   Furthermore, following the recommendation of the referee we have added the following sentence in the discussion: "In this study we have not investigated the impact of the rotational speed of the rotor and the blade pitch angles on up- and downwind conditions, which could be a subject of future work".

3. You should check the whole text for clarity, and to correct the text in respect of editorial guidelines. For example, write 'Figure' and not 'Fig.' at the beginning of a sentence. When you present a figure for the first time, make a dedicated sentence to present it (axes, colors, curves, etc): 'Figure X presents...'. Other examples are listed below.

   We have checked the text of the manuscript and edited according to the recommendation of the referee.

**1.3 Minor comments**

1. L19: Reformulate sentence. What do you mean by a 'good wind resource' ?

   The sentence "the generally good wind resource" is written as "the generally strong wind"

2. L19-21: I suggest that you reformulate the sentence

   In the direction of the reducing the length of the article we removed this sentence.

3. L84: Is 1 Hz enough ? Why ?

   The dynamic response of this floating wind turbine, as far as it concerns the yaw, pitch and roll motions, is mainly characterized by fluctuations that take place in frequencies lower than 0.1 Hz. This is presented in the work of Jacobsen and Godvik (2021) and also shown in Fig. B1 of this manuscript. For this reason we think that the 1 Hz sampling rate was adequate for the needs of this study.

4. L89: Do you know why there is a 3% decrease ?

   The pitch angle is proportional to the restoring torque due to buoyancy, which in turn is proportional to the thrust exerted on the wind turbine from the wind. The thrust is proportional to the product of the square of the wind speed and the thrust coefficient. A decrease of the thrust coefficient above the rates speed leads to a decrease in the thrust and thus to a decrease of the pitch angle.

   The following sentence was added in the text: "The decrease to $3°$, that is observed when the wind is higher that the rotor rated speed, is attributed to a decrease in the thrust exerted on the wind turbine's rotor, due to the pitch of the wind turbine's blades."

5. L102: Please clarify 'area covered by the rotor'

   We have rewritten this part as "swept area of the wind turbine"

6. L109: After the end of the sentence is probably the best place time to explain how long it takes to make a full PPI scan, as we don't want this important information to be buried later down in the text

   We have added the information of the duration of the PPI scan at the end of the sentence

7. L122: I wonder if this paragraph should be in this article. You only show this in the appendix. Also, is the HS2 the same as HS4 ?

   We have move the last paragraph of Sect. 2.1 to Appendix A. All the five wind turbines of the Hywind Scotland are of the same type. This is stated in the second sentence of Sect. 2.

8. Figure 3: The orange curve is not very clear. Maybe select a more visible color ?

   We removed the orange line, and instead we have colored the dots at the range where adjacent wind turbines could be located with gray.

9. Sect. 2.2.1: Can you clarify if you used the SNR to filter data ? Did you use a SNR threshold ?

   For the filtering of the *Wind Iris* data we used a parameter that is provided by the instrument. The parameter flags whether a radial wind speed estimation is "good" or "bad" according to the manufacturer. We don't know exactly how this parameter is calculated but most probably includes the SNR. We re-wrote the sentence as: "This data was selected based on the *Radial Wind Speed Status index* (*RWS Status index*), a parameter that is provided by the *Wind Iris*, which describes the quality of the radial wind speed estimation of each measurement (for more information we refer to the *Wind Iris* user manual (Avent Lidar Technology, Version 2.1.1))"

10. Sect. 3.1: You present your assumptions. What about stationarity ? Do you assume that you are stationary in terms of incident wind speed conditions at 10 minutes ? At 2 hours ? I would appreciate that you mention stationary here, and later in the text.

    For the application of the upwind radial speed model it is not required the assumption of stationary time series of the wind. For this reason was not stated here. Stationarity is used as a criterion for the selection of the periods of interest in Sect 4.3.

11. L167: replace by 'weak at measurement height' and you should provide references for this assumption

    The text is replaced and the following reference is included at the end of the sentence Peña et al. (2009)

12. L179-180: Maybe discuss what possible effects you are neglecting. How could this be affected by attitude (pitch etc) ? I see you already refer to Borraccino et al. 2017 later, maybe try to link the sentences ?

We have investigated the induction for an inflow wind which is constant everywhere and where the presence of the ground, and the pitch of the turbine is neglected. We do that by the simple formula (2) (or (4)) which agrees with the work of Conway on the symmetry line. We have tried to use the full Conway solution which included deviations as you go away from the center line and also a small transverse and vertical component. This changed the derived induction factor marginally. We have also tried to mimic the presence of the ground (sea surface) by introducing a mirror image wind turbine under the surface and adding the two Conway solutions. No significant changes in the lidar-observed wind field were noted. What is missing, but is felt to be outside of the scope of this paper, is a detailed investigation of

- The impact of the tilt of the turbine on the flow in the induction zone.

- The impact of shear and veer in the inflow on the induction zone flow.

- Impact of inflow turbulence on the same

These effects are quite demanding to compute and we defer that to future work. However, it should be noted that we do not see strong correlation between veer, shear and the induction factor.

13. L258: Could you explain what you know of the stability on that day, so as to support your argument?

A limitation of our work is the absence of sea temperature measurements, that would enable an estimation of the atmospheric stability. The statement between the lines 258 and 260 is a hypothesis.

14. L264: You should explain the figure in the text. Also, please clarify this paragraph.

The paragraph: "To further investigate the relationship between the fitted values of the wind shear and veer and the resulting root-mean-square error, we plot the fitted values of $\frac{\partial \overline{u}}{\partial z}$ and $\frac{\partial \overline{v}}{\partial z}$ as a function of the mean horizontal wind speed and the corresponding value of $\varepsilon_u$ in Fig. 5. Negative wind shear cases are usually have $\varepsilon_u$ higher than 0.2 ms$^{-1}$. When $\varepsilon_u$ is lower than the value above, the wind shear is usually within $0 - 0.02$ s$^{-1}$, which can be considered as low values, typical of offshore conditions. However, an increasing trend of the wind shear values is found when the mean free wind speed is higher than 15 ms$^{-1}$. It is worth mentioning that among the estimated wind shear values with $\varepsilon_u < 0.02$ s$^{-1}$, 31% are negative. " is now rewritten as:

To further investigate the relationship between the fitted values of the wind shear and veer and the resulting root-mean-square error, we plot the fitted values of $\frac{\partial \overline{u}}{\partial z}$ and $\frac{\partial \overline{v}}{\partial z}$ as a function of the mean horizontal wind speed in Fig. 5. Each dot corresponds to a 10-minute period and the color highlights the corresponding value of $\varepsilon_u$. We observe that on average the wind shear is usually within $0 - 0.02$ s$^{-1}$, which can be considered as low values, typical of offshore conditions. However, an increasing trend of the wind shear values is found when the mean free wind speed is higher than 12 ms$^{-1}$. Furthermore, when $\varepsilon_u$ is usually higher than 0.2 ms$^{-1}$ then negative wind shear cases are found. This is observed in the 31% of the 10-minute periods examined.

15. L266: Maybe you should show a fitting function to help show the trend. As it is hard to know where all your points are, it would be interesting to statistically illustrate your trend (add a fitting function, and provide statistical values).

Following the recommendation of the reviewer we have updated the figure by including the mean shear and veer values in different 1-m/s wind speed bins. In addition we include error bars that correspond to the standard deviation of each mean value.

16. L271: You are presenting Fig. 6 in brackets at the end of the sentence. Figure 6 deserves better, and should be presented and described in a clearer manner. You may also remove the mention of the SCADA, as this has already been presented before.

The paragraph: "For assessing the accuracy of the model of Eq. 8, we compare the 10-minute mean horizontal wind speed at the hub height with the corresponding values of the nacelle-mounted anemometer, recorded in the SCADA system (see Fig. 6). For this purpose we use only cases where $\varepsilon_u < 0.2 \text{ ms}^{-1}$" is now rewritten as: "For assessing the accuracy of the model of Eq. 8, we compare the 10-minute mean horizontal wind speed at the hub height with the corresponding values of the nacelle-mounted anemometer. Figure 6(a) presents a scatter plot between the estimated free wind at hub height based on the *Wind Iris* and the nacelle anemometer data. For this purpose we use only cases where $\varepsilon_u < 0.2 \text{ ms}^{-1}$."

17. L280-282: This carries an important message. This deserves to be made more obvious, and move out of the middle of the paragraph.

This sentence is now moved at the beginning of the paragraph and the whole paragraph was edited accordingly.

18. L292: I suggest that you add references on actuator disk models.

The following reference is added at the end of the sentence "(Hansen, 2015, Ch. 4)"

19. L295: Please clarify sentence, as I do not understand it. From what I currently understand, it seems that the non-correlation is explained by the fact that they may be correlated.

We have simplified the sentence to the following:
"Overall, we do not observe any correlation between the induction factor and the estimated wind speed shear or veer (plots not shown)". To support our answer to the reviewer we have included Fig. 1. The figure presents scatter plots of the induction factor values versus the shear (top row) and veer (bottom row) for two different mean wind speeds, below (i.e. 8 m/s) and above (i.e.14 m/s) rated speed. The plots present the absence of correlation between these parameters.

20. L306: Maybe you should explain why TI is different for similar U10 wind speeds ?

We have added the following sentence in the article: "The different TI levels could be attributed to the atmospheric stability conditions and possibly to the sea state"

21. L315: Please specify in what figure I can see this wind shear increase etc

The sentence is re-written as: "However, we observe that the wake deficit is still visible but less strong when both the TI and the wind shear increase, as seen in Fig. 8(d)."

[Figure]

**Figure 1.** Scatter plot of the induction factor versus the wind shear and veer for two different mean wind speeds at hub height, corresponding to the below (left colum) and above (right column) rated range.

22. L325: As of this paragraph, you use 30 minute, 2 minute, and 10 minute averages. Could you clarify this choice ? Maybe you can detail that you will be doing this a bit earlier, such as at the beginning of the section. It is quite hard to follow when you bring new averaging types at the beginning of each paragraph without prior notice.

The 2-minute period corresponds to the time needed for the completion of one PPI scan from the *Galion* lidar. During this period an averaging takes place only in those cases where more than one measurements is found in a grid cell. The 10-minute period was selected to estimate the free wind characteristics. Finally the 30-minute averaging was chosen for statistical purposes so as to increase the available data set for the determination of the mean wake characteristics. In the

revised version of our manuscript we have restructured Sect. 4.3 in a way that we think clarifies the reasons why different averaging times were used in the study.

23. 321-324: Could you provide some clearer explanation of why Fig. 8D does not show these 'stripes' ?

We are not sure why the stripes that observed in Fig 8(a) are not present in Figs. 8(b)-(d). Our hypothesis is that this could be attributed to either or both the atmospheric stability and variations of the wind direction. We have to note here that in Fig. 8(a) we observe the highest value of the transverse wind component among the four cases presented in Fig. 8. In the updated version of the manuscript we have re-written the sentence: "This feature may be artificially created by the flow characteristics and the relative slow scanning speed of the wind lidar." as: "This feature could be attributed to a combination of the flow characteristics (i.e. atmospheric stability and/or wind direction changes) and the relative slow scanning speed of the wind lidar."

24. Figure 9: I wonder how useful this figure is to the discussion.

We agree with the reviewer that this plot could be omitted from the article. Following the recommendation of the reviewer to reduce the size (and the number of figures) in the article, we have decided to remove it from the revised version.

25. Figure 10: Maybe consider showing the slices (such as in Fig. 10c) for other x/D values ? I also realize that you do not refer to Fig. 10D in the text. If you don't mention it, then consider removing it. Finally, maybe you can remind what LiDAR you used in the caption of the figure.

Fig.10(d) was briefly discussed in the lines 355 and 356 of the original version, which stated "The standard deviation of the radial wind speed reveals two bands of high turbulence at the edges of the wake, which correspond to the areas where the wake mixed with the free flow". Since the main topic of the article is the mean flow and following the recommendation of the reviewer to reduce the number of figures with have removed Figure 10D and we have added two more figures that show the wake profile in two more downstream distances. We have chosen the distance 2.7D and 8.3D which correspond to the lower and upper limit of the wake range examined here. We have also re-written the caption of the figure in order to add the information of the lidar used.

26. Figures 11 and 12: They do not serve much in the discussion, maybe they should be removed ? However, I think that Figure 11 is quite interesting, and carries a more interesting and simpler message that Figure 14 that could be removed instead, and replaced with a couple of sentences.

Figure 11 shows the agreement between Eq. 9 and the wake measurements. We think that it is important because it supports our choice of the downwind distances that were selected in the analysis. Figure 11 is discussed between the lines 360 and 367 (line numbers correspond to the original version) and we have kept this text also in the revised manuscript. As far as it concerns Figs. 12 and 14, following the suggestion of the reviewer to reduce the number of figures, we have removed them from the revised manuscript.

27. Figure 13: I don't see the values at the higher end of the x/D range (near 9.6 in yellow), but they should appear on the front.

215 There was an error in the color scaling of the plot in the original version of the article. It is corrected in the revised version of the manuscript.

28. Figure 14: Also present the vertical axis

Following the comment #26 of the reviewer we have removed Figure 14.

29. Figure 15: The label should specify 'horizontal velocity' ?

220 The velocity profiles in Fig. 15 correspond to the radial wind speeds. We therefore added the word "radial" in the label.

30. Figure 16: You show the constant du/dz and dv/dz, which you assume at the beginning of the article. Maybe you should make this clear, as this is quite a nice result.

We have added the following sentence: "Overall, we observe that the measured longitudinal and transverse wind components increase linearly with height, which supports our hypothesis of constant shear and veer in Eq. X."

225 31. L430: You should consider reminding the reader how you are detecting the wake center. Also make a proper presentation of Fig. 17.

We have added the following sentence in Sect. 4.7.1: "The $y_0$ values are derived from fitting the radial speed model of Eq. 9 to the measurements of the scanning wind lidar in different downwind distances. Subsequently, the mean (dots) and the standard deviation (error bars) of the $y_0$ values for each of the cases examined, were calculated."

230 Furthermore, we have added the following sentence in the *Discussion*: "As a future best practice, we recommend a combination of PPI and RHI scanning configurations to study the mean characteristics of floating wind turbine wakes. For example this would enable a more thorough study of the features that we observed in the downwind propagation of the wake center (Fig. 14)"

32. L434: Where does this 2° value come from ?

235 It comes from the values presented in the fourth column of Table 1. We rewrote the sentence as following: "According to those values the maximum yaw misalignment of the examined cases was equal to -2°, which could result in a transverse displacement by -0.3$D$ at 8$D$"

33. L461: what is 'a' in the equation ?

It corresponds to the induction factor. We have added the following text here: "..equal to twice the induction factor 240 $2a \times 100\%$ ...

**1.4 Typos etc:**

1. L8: Replace by 'along the horizontal plane' ?

   We replaced the sentence: "The wake flow is measured by a wind lidar scanning in a horizontal plan position indicator mode," with "The wake flow is measured along a horizontal plane by a wind lidar scanning in a plan position indicator mode,".

2. L23: replace by 'realistically model', or '...flows realistically'

   Replaced by "realistically model"

3. L31: Add 'For example, the results...'

   Suggestion added

4. L32: remove 'for example'

   Words are removed

5. L32: remove 'to' by 'in', add comma after 'recovery'

   The corrections are done

6. L39: 'Focused'

   Corrected

7. L45: Replace 'enhancing' by 'increasing'

   Replaced

8. Figure 1 label: Replace 'whose y-axis' by 'where the y-axis'

   Replaced

9. L72: Remove 'have been'

   Removed

10. L80: Replace 'relative to that' by 'of the'

    Replaced

11. L81: End sentence at nacelle. New sentence: 'The MRU measured the rotation...'

    The sentence was corrected following the suggestion of the reviewer

12. L86: Replace 'Section' by 'Sect.'

    Replaced

13. L87: New paragraph after 'longitudinal'. And the longitudinal what ? Longitudinal axis ?

The sentence was corrected following the suggestion of the reviewer

14. L95: add 'used for this study were the Wind Iris' …

Suggestion added

15. L101: Replace 'Leveled' by 'Level'

Here we decided to delete the word "horizontally" and replace the word "leveled" with "levelled".

16. L103: Replace 'points' by 'dots' . Replace 'defining' by 'that define'

Replaced

17. L109: replace 'spanned from' by 'spanning'

Replaced

18. Sect. 2.2.2: Maybe add something like 'As described in Sect. 4.3, the Galion will be filtered for cases where the scans are horizontal'.

The sentence "In the case of the *G4000 Galion*, the data filtering was based on the *intensity* values provided by the *Galion* software (SgurrEnergy Ltd., 2017),..." in the original manuscript is re-written as:

As described in Sect. 4.3, the *G400 Galion* wind lidar data was filtered for cases where the PPI scans were leveled. The data filtering was based on the *intensity* values provided by the *Galion* software (SgurrEnergy Ltd., 2017),

19. L153: Replace 'upwind and wake conditions' by 'upwind and downwind' ?

Replaced

20. L171: Remove 'Further'

Removed

21. L172: A repetition, as you also define the induction factor alpha later (L177).

"We have removed the text "$a$ is the induction factor which depends on the wind operation" from the updated version.

22. L191: Replace 'i=1,2,3 and 4' by 'i=1,2,3,4' ?

Replaced

23. Sect. 3.2: Maybe it would be interesting to present why you want a radial speed model. What is you objective here ?

A Doppler lidar acquires radial speed measurements. We want to use a model to express the Doppler lidar measurements as a function of the mean wake flow parameters. We think that we present this in the first three sentences of the Sect. 3.2.

24. L221: I suggest a new paragraph after 'deficit', and write 'We assume, using Equation 9...'

The suggestion is implemented in the revised manuscript

25. L228: I suggest 'The solver was applied in each streamwise distance, and the measurements at x/D = 6 were chosen as an input...'

The suggestion is implemented in the revised manuscript

26. L229: Suggest 'This choice is supported by the sufficient number of measurements'

The suggestion is implemented in the revised manuscript

27. L234: Suggest 'a total of 10529' and remove 'in total' later

The suggestion is implemented in the revised manuscript

28. L236: End sentence at 'periods'. Begin new sentence with 'They were selected'

The suggestion is implemented in the revised manuscript

29. L241: I suggest a new paragraph here

The suggestion is implemented in the revised manuscript

30. Figure 4: Consider moving the figure to this page

Figure moved

31. L277: Suggest replacing 'selection' by 'choice'

Replaced

32. L293: Replace 'gets larger than' by 'exceeds'

Replaced

33. L318-319: Replace with 1D, 2D, 4D

Replaced

34. L324: Provide reference to figure instead of 'In the plot'

The text "In the plot," is replaced by "In Fig.8(d)"

35. L325: Provide reference to the 'selected data', which is this selected data ?

The sentence: "The selected data were gathered in 30-minute periods and averaged to produce 170 cases of a time-averaged wake in a fixed frame of reference" is rewritten as: "The individual single scans that were acquired during 10-minute periods that satisfied the criteria stated in Sect. 4.3 were gathered in 30-minute periods and averaged to produce 170 cases of a time-averaged wake in a fixed frame of reference"

36. L394: You should properly introduce Fig. 13

   The sentence: "The results using Eq. 12 and Eq.3 in Eq. 10 are presented in Fig. 11 for a 30-minute period with a mean free speed equal to 13 ms$^{-1}$. We find that the distribution of the velocity profile of the wake along the lateral axis to the wake propagation can be considered self-similar for distances between $2.2 < x/D < 9.6$. We determine this visually, since the estimated profiles of the velocity defect $f$ can be expressed as a function of $\xi$ (Pope, 2000, Ch. 5)" is now rewritten: "An example of using Eqs. 12 and 13 in Eq. 10 is presented in Fig. 11 for a 30-minute period with a mean free speed equal to 13 ms$^{-1}$. The figures presents the transverse profile of the self-similar velocity defect for different downwind distances between $2.7 < x/D < 8.3$. We visually observe that the estimated profiles of the velocity defect $f$ can be expressed as a function of the variable $\xi$ of Eq. 10."

37. L397: Error with references

   We checked the references and updated the following Borraccino et al. (2017); Gryning and Floors (2019); Nanos et al. (2022); Porté-Agel et al. (2020); Wu and Porté-Agel (2012); Zhang et al. (2012).

**2  Answer to Referee 2**

Review of "Revealing inflow and wake conditions of a 6MW floating turbine" by Angelou et al. provides details of the first ever wake measurements on a floating offshore wind turbine. This has generated a lot of buzz in the research community and commend the authors for getting this timely paper out for review. Overall, I think this is a very interesting paper and provides a lot of interesting physics on how the wake propagates offshore on a floating offshore wind turbine. The authors mainly summarize that the transverse profile of the wake can be described by a self-similar wind speed deficit, following a Guassian distribution and support the hypothesis that the TI plays a major role in wake recovery. The theory proposed is sound, although the devil is in the details, and it would be good for the authors to clarify some of the below questions to make the paper clearer. We can't expect one paper to answer all our questions on wake propagation for a floating offshore wind turbine, but this is a great start, and this paper should be published after some of the comments are addressed. Reviewer #1 has provided an extensive list of comments on the paper, some which I also share so will not duplicate them here.

**3  Major Comments**

1. Variations of 1 degree of the nacelle for the scanning lidar data can be significant, can result in +/- 25 m height difference in beams at a range of 1500 m. Yes, for the structure of that size itself the stability is great, but for the scanning lidars, it might not be ideal, unless active motion compensation was performed. There is some preliminary analysis done for one case study, but a more thorough assessment on how this motion would affect the results/conclusions would be helpful.

   We agree with the comment of the reviewer regarding the stability of the floating wind turbine and it is impact on the measuring locations of the scanning wind lidar. Ideally, as the reviewer is noting in the case of a floating structure, an active motion compensation should be implemented during the measuring campaign. However, given the fact that the floating wind turbine in this study was experiencing relative low roll and pitch rotations, as well as that our investigation focused on downwind ranges down to $8.3D$, we do not expect that the vertical variability had a significant impact to the study of the wake. However we have added the following sentence in the *Discussion*: " Furthermore, due to the relative stable response of the floating wind turbine examined in this study, an active motion compensation was not implemented in the scanning wind lidar. However, this could be necessary for the monitoring of the wake flow, especially in the far wake region, in the case of floating wind turbines that are characterized by larger motions.".

2. Can you mention about the turbine blade pitch angles, tip speed ratios etc. during these conditions? How would have the blade pitching effect the wake here? Does the blade pitching only happen above rated winds like a typical wind turbine or is the motion of the turbine coupled with blade pitching somehow?

   Please see our answer to the comment 1 in the *Minor questions* subsection in the next page.

3. What were the sea state conditions during the study period or cases analyzed? Do you have any sea surface temperature measurements for estimating atmospheric stability along with air temperature measurements (say using a Bulk Richardson number estimates)?

Unfortunately, we do not have information about the atmospheric stability, since at the time of the experiment we did not have measurements of the sea temperature. We highlighted this as one of the things that it was missing from the study in the discussion of the original version. In the updated version we re-wrote that part as: "A limitation of this study was that the lack of information about atmospheric stratification. Unfortunately, during the lidar measuring campaign, there were not available any sea surface temperature measurements, which would enable the characterization of the atmospheric stability. This knowledge would allow a more thorough investigation of the wake properties and their dependence on atmospheric conditions.".

4. Making the conclusions very specific to this type of wind turbine would be important. As the wake recovery can also depend on the control strategy (not just the blades but the entire structure) and type of mooring of the floating offshore wind turbine, which is not discussed here. So would recommend stressing that this result is not generic for all floating offshore wind turbines.

We agree with the comment of the reviewer. We tried in the first paper to emphasize that in this wake study the floating wind turbine was very stable, which may not be the case for other floating wind turbines. When the motions of the floating wind turbine are similar to the ones examined here then we should expect a similar trend as the one presented in this article.

We re-wrote the last sentence of the abstract as: "Furthermore, we do not observe any additional spread of the wake due to the motion of the floating wind turbine examined in this study".

Furthermore, we have added the following sentence in the *Conclusions*: "These results indicate that the wakes of floating wind turbines, that do not experience high pitch and roll rotations, will have similar characteristics to those of fixed wind turbines."

**3.1 Minor questions**

1. Figure 2: Why is the pitch angle reducing post rated wind speeds? Is there any active control of the floating structure? It was not clear from the paper.

The pitch angle is proportional to the restoring torque due to buoyancy, which in turn is proportional to the thrust exerted on the wind turbine from the wind. The thrust is proportional to the product of the square of the wind speed and the thrust coefficient. The decrease of the pitch angle is due to the decrease of the thrust coefficient, at the above rated speed.

Regarding the operation of the wind turbine, we have added the following sentence in Section 2: "The blade pitch of the turbine is controlled as a conventional bottom fixed wind turbine during operation below rated wind speed. However, above the rated wind speed the blade pitch control system interacts with a floater motion control system."

2. Why was 0.32 Hz chosen for the scanning lidar time averaging?

The 0.32 Hz was chosen in order to ensure high data availability throughout the measuring range, while moving the scanner head in a generally slow speed that minimized potential failure of the Doppler lidar. We have added the following sentence in the text: "The 0.32 Hz was chosen in order to ensure high data availability along the measuring range."

3. Appendix B has no text but just a figure. Can you please provide some explanation here?

We have added the following text in Appendix B: "Figure B1 presents the power spectral density of the pitch (a) and roll (b) angles. We observe that in general higher energy levels appear in the spectrum of the pitch angle which corresponds to oscillations longitudinal to the yaw direction. However, in the roll angles we observed a more predominant frequency peak at 0.01 Hz and 0.03 Hz, which correspond to the natural frequency of the sway and roll motions of the wind turbine Jacobsen and Godvik (2021), respectively."

4. There are some formatting issues, so would recommend authors to carefully check the figures are being referenced and discussed. Also, noted some consistency issues in the paper, hoping all these can be sorted out prior to the second round of revisions.

Following the suggestion of the *Reviewer 1*: i. we have decreased the length of the article by removing Figs. 9,10(b) % (d), 12 and 14, ii and we have restructured Sects. 4.3 – 4.7 of the origal version.

**References**

Avent Lidar Technology: Wind Iris, User manual, Leosphere Group, Version 2.1.1.

Borraccino, A., Schlipf, D., Haizmann, F., and Wagner, R.: Wind field reconstruction from nacelle-mounted lidar short-range measurements, Wind Energy Science, 2, 269–283, https://doi.org/10.5194/wes-2-269-2017, 2017.

Gryning, S. E. and Floors, R.: Carrier-to-noise-threshold filtering on off-shore wind lidar measurements, Sensors, 19, https://doi.org/10.3390/s19030592, 2019.

Hansen, M. O.: Aerodynamics of Wind Turbines, Routledge, 3rd edn., https://doi.org/10.4324/9781315769981, 2015.

Jacobsen, A. and Godvik, M.: Influence of wakes and atmospheric stability on the floater responses of the Hywind Scotland wind turbines, Wind Energy, 24, 149–161, https://doi.org/10.1002/we.2563, 2021.

Nanos, E. M., Bottasso, C. L., Manolas, D. I., and Riziotis, V. A.: Vertical wake deflection for floating wind turbines by differential ballast control, Wind Energy Science, 7, 1641–1660, https://doi.org/10.5194/wes-7-1641-2022, 2022.

Peña, A., Hasager, C. B., Gryning, S.-E., Courtney, M., Antoniou, I., and Mikkelsen, T.: Offshore wind profiling using light detection and ranging measurements, Wind Energy, 12, 105–124, https://doi.org/https://doi.org/10.1002/we.283, 2009.

Pope, S. B.: Turbulent Flows, Cambridge University Press, https://doi.org/10.1017/CBO9780511840531, 2000.

Porté-Agel, F., Bastankhah, M., and Shamsoddin, S.: Wind-Turbine and Wind-Farm Flows: A Review, Boundary-Layer Meteorology, 174, 1–59, https://doi.org/10.1007/s10546-019-00473-0, 2020.

SgurrEnergy Ltd.: Galion Toolbox, User manual Revision 2017 B3, 2017.

Wu, Y.-T. and Porté-Agel, F.: Atmospheric turbulence effects on wind-turbine wakes: An LES study, Energies, 5, 5340–5362, https://doi.org/10.3390/en5125340, 2012.

Zhang, W., Markfort, C. D., and Porté-Agel, F.: Near-wake flow structure downwind of a wind turbine in a turbulent boundary layer, Experiments in Fluids, 52, 1219–1235, https://doi.org/10.1007/s00348-011-1250-8, 2012.